# Gaussian process surrogate models for neural networks

## Abstract

The lack of insight into deep learning systems hinders their systematic design. In science and engineering, *modeling* is a methodology used to understand complex systems whose internal processes are opaque. Modeling replaces a complex system with a simpler surrogate that is more amenable to interpretation. Drawing inspiration from this, we construct a class of surrogate models for neural networks using Gaussian processes. Rather than deriving the kernels for certain limiting cases of neural networks, we learn the kernels of the Gaussian process empirically from the naturalistic behavior of neural networks. We first evaluate our approach with two case studies inspired by previous theoretical studies of neural network behavior in which we capture neural network preferences for learning low frequencies and identify pathological behavior in deep neural networks. In two further practical case studies, we use the learned kernel to predict the generalization properties of neural networks.

## 1 Introduction

Deep learning systems are ubiquitous in machine learning but sometimes exhibit unpredictable and often undesirable behavior when deployed in real-world applications (Geirhos, Jacobsen, et al. 2020; D'Amour et al. 2020). This gap between idealized and real-world performance stems from a lack of principles guiding the design of deep learning systems. Instead, deep learning practitioners often rely upon a set of heuristic design decisions that are inadequately tied to a system's behavior (Dehghani et al. 2021), driving calls for explainability, transparency, and interpretability of deep learning systems (Lipton 2016; Doshi-Velez and Kim 2017; Samek et al. 2017) especially as these systems are more widely applied in everyday life (Bommasani et al. 2021).

Machine learning is not unique in seeking to understand a complex system whose inputs and outputs are observable but whose internal processes are opaque—this challenge occurs across the empirical sciences and engineering. An explanatory tool that is foundational across these disciplines is that of *modeling*, that is, representing a complex and opaque system with a simpler one that is more amenable to interpretation.[1] Modeling makes precise assumptions about how a system may operate while abstracting away details that are irrelevant for a given level of understanding or a given downstream use case. These properties are valuable for a framework for understanding deep learning as they are in other scientific and engineering disciplines.

As the popularity of deep learning has grown, a number of proposals have been made for modeling these systems. Numerous mathematical models of deep learning have been developed (Roberts et al. 2022), and some surprising phenomena, such as adversarial examples (Szegedy et al. 2014), have been captured with a mathematical analysis (Ilyas, Santurkar, et al. 2019). However, existing mathematical models, which are limited to well-understood mathematical tools, are unable to capture the properties of machine learning systems as applied in practice (Nakkiran 2021). Beyond mathematical models, localized models have aimed to explain the predictions of machine learning systems on a per-example basis (Ribeiro et al. 2016; Koh and P. Liang 2017; Zhou et al. 2022; Ilyas, Park, et al. 2022), but these approaches are, by construction, only partial explanations of the behavior of the end-to-end system.

What might an alternative modeling approach—one that captures salient aspects of applied systems in a global fashion—look like? We appeal to two domains for inspiration. In engineering design, *surrogate*

---

[1]Though some architectural components of a deep learning system are commonly referred to as a *model*—as in "neural network model"—we use *modeling* to refer to the methodology of idealizing a complex system as a simpler one.

*models* (G. G. Wang and Shan 2006) emulate the input-output behavior of a complex physical system, allowing practitioners to simulate effects that are consequential for design or analysis without relying on costly or otherwise prohibitive queries from the system itself. In cognitive science, *cognitive models* (Sun 2008; McClelland 2009) describe how unobservable mental processes such as memory or attention produce the range of people's observed behaviors. Both domains abstract away internal details, such as real-world constraints on a physical system or neural circuitry in the brain, instead treating the target process or system as a *black box*. At the same time, both surrogate and cognitive models are constructed to replicate the end-to-end behavior of the target system and thus are complete where localized explanations are not.

We explore an analogous approach to investigate deep learning systems by constructing *surrogate models for neural networks*. We first must choose an appropriate family of surrogate models. Gaussian processes (GPs) are a natural choice, with appealing theoretical properties specific to the study of neural networks (NNs); namely, certain limiting cases of NN architectures are realizable as GPs (Neal 1996; Li and Y. Liang 2018; Jacot et al. 2018; Allen-Zhu et al. 2019; Du et al. 2019). However, in contrast to these analytic approaches, we aim to explore the scientific and practical utility of idealizing NNs with GPs using a *data-driven* approach to estimating the kernel functions. Separately, the learned kernel of a GP is often interpretable (Wilson and Adams 2013); we use this fact to study the prior over functions represented by a GP that accounts for observed neural network behavior in less-restricted settings. With this approach, we capture a number of known phenomena, including a bias towards low frequencies and pathological behavior at initialization, in a cohesive framework. Finally, we demonstrate the practical benefits of this framework by predicting the generalization behavior of models in an NN family.

## 2 Background

In **surrogate modeling**, we approximate a complex black-box function with a simpler surrogate model that is more amenable to interpretation. Surrogate models have many applications: In optimization, they are often used to approximate queries from expensive-to-evaluate functions (Snoek et al. 2012; Shahriari et al. 2016; Xue et al. 2020); in other applications, surrogate models have been used to gain insight into large physical systems, such as the global fluxes of energy and heat over the earth's surface (Camps-Valls et al. 2015).

**Cognitive models** have been used by cognitive scientists since the 1950s to gain insight into another black box—the human mind (Newell et al. 1958). Bayesian models of cognition, in particular, offer a way to describe the inductive biases of learning systems in the form of a prior distribution (Griffiths et al. 2010). As deep NNs have become more prevalent in machine learning, researchers have started to use methodologies from cognitive science to interrogate otherwise opaque models (Ritter et al. 2017; Geirhos, Rubisch, et al. 2019; Hawkins et al. 2020). The success of these efforts suggests that other methods from cognitive science—namely, cognitive modeling—may be applicable to machine learning systems.

**Gaussian processes** (GPs; Carl E. Rasmussen and Williams 2006) are probabilistic models that specify a distribution over functions. A GP models any *finite* set of $N$ observations as a multivariate Gaussian distribution on $\mathbb{R}^D$, where the $n$th point is interpreted as the function value, $f(\mathbf{x}_n)$, at the input point $\mathbf{x}_n$. GPs are fully characterized by a mean function $m(\mathbf{x})$, usually taken to be degenerate as $m(\mathbf{x}) = \mathbf{0}, \forall \mathbf{x}$, and a positive-definite kernel function $k(\mathbf{x}, \mathbf{x}')$ that gives the covariance between $f(\mathbf{x})$ and $f(\mathbf{x}')$ as a function of $\mathbf{x}$ and $\mathbf{x}'$.

Formally, let $\mathbf{X}$ be a matrix of inputs and $\mathbf{y}$ be a vector of output responses. Due to the marginalization properties of the Gaussian distribution, the posterior predictive distribution of a GP for a new input $\mathbf{x}_*$, conditioned on dataset $\mathcal{D} = \{\mathbf{X}, \mathbf{y}\}$ and assuming centered Gaussian observation noise with variance $\sigma^2$, is Gaussian with closed-form expressions for the mean and variance:

$$\mathbb{E}[f(\mathbf{x}_*) \mid \mathcal{D}] = m(\mathbf{x}_*) + \mathbf{k}_*^T (\mathbf{K} + \sigma^2 \mathbf{I})^{-1} (\mathbf{y} - m(\mathbf{x}_*)) \tag{1}$$

$$\mathbb{V}[f(\mathbf{x}_*) \mid \mathcal{D}] = k(\mathbf{x}_*, \mathbf{x}_*) - \mathbf{k}_*^T (\mathbf{K} + \sigma^2 \mathbf{I})^{-1} \mathbf{k}_* \tag{2}$$

where $\mathbf{K}$ is the $N \times N$ Gram matrix of pairwise covariances, $k(\mathbf{x}_i, \mathbf{x}_j)$, and $\mathbf{k}_* = [k(\mathbf{x}_1, \mathbf{x}_*), \ldots, k(\mathbf{x}_N, \mathbf{x}_*)]^T$.

The kernel function $k$ specifies the prior on what kind of functions might be represented in observed data; for example, it can express expectations about smoothness or periodicity. Parametric kernels have hyperparam-

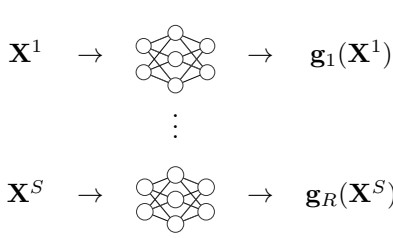

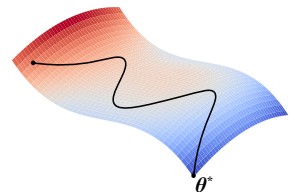

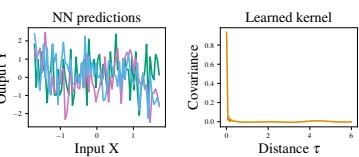

**Step 1**: Collect predictions across models $\mathbf{g}_r$ and across target functions (datasets) $\mathbf{X}_s$.

**Step 2**: Fit Gaussian process hyperparameters $\theta$ to the aggregate predictions via Objective (**6**).

**Step 3**: Analyze the kernel learned from the aggregate predictions. Here, the learned kernel reveals the quickly varying behavior of particular neural networks.

**Figure 1: Outline of the surrogate modeling approach.** We learn a Gaussian process surrogate model for a neural network family applied to a task family by learning kernel hyperparameters from aggregated neural network predictions across datasets. We interpret the learned kernel to derive insights into the properties of the neural network family; for example, biases towards particular frequencies (see Section (**4.1**)), or expected generalization behavior on a new dataset (see Section (**4.4**)).

eters $\theta$ that affect this prior and thus the posterior predictive. These kernel hyperparameters can be adapted to the properties of a dataset, thus defining a prior over functions that is appropriate for that context. GP kernel hyperparameters are typically learned via gradient-based optimization to maximize the GP marginal likelihood, $p(\mathbf{y} \mid \mathbf{X})$. Again due to properties of the GP, this marginal likelihood has the closed-form expression:

$$\log p(\mathbf{y} \mid \mathbf{X}) = -\frac{1}{2}\mathbf{y}^T \left(\mathbf{K}_\theta + \sigma_n^2 I\right)^{-1} \mathbf{y} - \frac{1}{2}\log|\mathbf{K}_\theta + \sigma_n^2 I| - \frac{n}{2}\log 2\pi \;, \tag{3}$$

We write the Gram matrix as $\mathbf{K}_\theta$ to indicate that it depends on kernel hyperparameters via a particular parameterization. In this work, we make use of two kernel parameterizations: the **Matérn kernel** (MK; Matérn 1960) and the **spectral mixture kernel** (SMK; Wilson and Adams 2013). Specifically, following Snoek et al. (2012), we use the automatic relevance determination (ARD) 5/2 MK, given by:

$$k(\mathbf{x}, \mathbf{x}') = \theta_0 \left(1 + \sqrt{5r^2(\mathbf{x}, \mathbf{x}')} + \tfrac{5}{3}r^2(\mathbf{x}, \mathbf{x}')\right)\exp\left\{-5\sqrt{5r^2(\mathbf{x}, \mathbf{x}')}\right\} \qquad r^2(\mathbf{x}, \mathbf{x}') = \sum_{d=1}^{D}(x_d - x_d')^2/\theta_d^2 \;, \tag{4}$$

where each $\theta_d$ is the lengthscale parameter for dimension $d$, which captures how smoothly the function varies along that dimension. The SMK is derived by modeling the spectral density of a kernel as a scale-location mixture of Gaussians and computing the Fourier transform of the mixture (Wilson and Adams 2013), giving:

$$k(\tau) = \sum_{q=1}^{Q} w_q \cos\left(2\pi^2 \tau^T \mu_q\right) \prod_{p=1}^{P} \exp\left\{-2\pi^2 \tau_p^2 v_q^{(p)}\right\} \;. \tag{5}$$

Here, $k(\tau)$ gives the covariance between function values $f(\mathbf{x})$ and $f(\mathbf{x}')$ whose corresponding input values $\mathbf{x}$ and $\mathbf{x}'$ are a distance $\tau$ apart. For a $Q$-component spectral mixture, $w = \{w_i\}_{i=1}^{Q}$ are scalar mixture weights, and $\mu_i \in \mathbb{R}^P$ and $v_i \in \mathbb{R}^P$ are component-wise Gaussian means and variances, respectively. Appendix (**A**) details how the hyperparameters of the MK and the SMK control the respective priors on functions.

## 3 Learning a Gaussian process surrogate model from neural network predictions

In this section, we detail the goals and approach of the surrogate modeling framework. In brief, our approach involves collecting neural network predictions across a set of neural network models and across a set of datasets, and estimating GP kernel hyperparameters from these predictions by maximizing the marginal likelihood across model-and-dataset pairs; see Fig. (**1**) for a schematic.

### 3.1 Formal framework

Our goal is to capture shared properties among a family of neural networks models $\mathfrak{F}$ as applied to a family of datasets $\mathfrak{D}$. Here, a model family $\mathfrak{F}$ is a set of neural networks $\{\mathbf{g}_0, \ldots, \mathbf{g}_R\}$ that share in design

choices (*e.g.,* architecture, training procedure, random initialization scheme) but differ in quantities that are randomized prior to or during training (*e.g.,* parameter initializations).[2] Similarly, a dataset family $\mathfrak{D}$ is a set of datasets $\{\mathcal{D}_0, \ldots, \mathcal{D}_S\}$ that share some underlying structure as in multi-task and meta-learning settings (Caruana 1997; Hospedales et al. 2020). We consider supervised learning, in which each dataset consists of inputs and targets, $\mathcal{D} = (\mathbf{X}, \mathbf{y})$. Importantly, we fit surrogate model parameters $\theta$ to a *behavioral dataset* of the model family evaluated on the dataset family, and not the ground truth datasets themselves.

**Data.** We construct a component of the surrogate model training dataset as follows: We sample a model index $r$ and a dataset index $s$. The corresponding dataset is split into a training set and an evaluation set, $\mathcal{D}_s = \mathcal{D}_s^{\text{train}} \cup \mathcal{D}_s^{\text{eval}}$. The corresponding model $\mathbf{g}_r$ is fit the training set $\mathcal{D}_s^{\text{train}} = (\mathbf{X}_s^{\text{train}}, \mathbf{y}_s^{\text{train}})$ according to the training procedure specified by the choice of model family $\mathfrak{F}$, producing $\mathbf{g}_r^{\text{fit}}$. We then collect the predictions of the trained model on the evaluation set, $\mathbf{g}_r^{\text{fit}}(\mathbf{X}_s^{\text{eval}})$, to produce the component $(\mathbf{X}_s^{\text{eval}}, \mathbf{g}_r^{\text{fit}}(\mathbf{X}_s^{\text{eval}}))$ consisting of the *ground truth inputs* paired with the *neural network behavioral targets* from the evaluation set. We aggregate the ground truth inputs and the neural network behavioral targets across pairs to produce the *surrogate model training dataset*, $\left((\mathbf{X}_{\mathbf{s_1}}^{\text{eval}}, \mathbf{g}_{r_1}^{\text{fit}}(\mathbf{X}_{s_1}^{\text{eval}})), \ldots, (\mathbf{X}_{\mathbf{s_T}}^{\text{eval}}, \mathbf{g}_{r_T}^{\text{fit}}(\mathbf{X}_{s_T}^{\text{eval}}))\right)$.

---

**Algorithm 1:** Training and evaluation of the GP surrogate model described in Section (**3**).

**hyperparameters:** model family $\mathfrak{F}$,
dataset family $\mathfrak{D}$,
model-dataset count $T$,
GP parameterization $\theta$

// Step 1 in Fig. (**1**)
**for** $t \in 1 \ldots T$ **do**
  Sample a model, $\mathbf{g}_{r_t} \sim \text{Unif}(\mathfrak{F})$
  Sample a dataset, $\mathcal{D}_{s_t} \sim \text{Unif}(\mathfrak{D})$
  Train the model, $\mathbf{g}_{r_t}^{\text{fit}} \leftarrow \text{train}(\mathbf{g}_{r_t}, \mathcal{D}_{s_t}^{\text{train}})$
  Evaluate $\mathbf{g}_{r_t}^{\text{fit}}(\mathcal{D}_{s_t}^{\text{eval}})$
**end**
// Step 2 in Fig. (**1**)
Optimize Objective (**6**) for $\theta^*$
// Step 3 in Fig. (**1**)
Analyze $\theta^*$ via $P_{\theta^*}$

---

**Surrogate model.** We fit the GP using type-II maximum likelihood estimation. Let $P_\theta(\mathbf{g}^{\text{fit}}(\mathbf{X}^{\text{eval}}) \mid \mathbf{X}^{\text{eval}})$ be the GP marginal likelihood of the dataset component $(\mathbf{X}^{\text{eval}}, \mathbf{g}^{\text{fit}}(\mathbf{X}^{\text{eval}}))$ under a GP with kernel hyperparameters $\theta$, as given in Eq. (3). We fit the surrogate model jointly across model-and-task pairs in the surrogate model training dataset by maximizing the joint marginal likelihood with respect to $\theta$:

$$\max_\theta \prod_{(r,s)} P_\theta(\mathbf{g}_r^{\text{fit}}(\mathbf{X}_s^{\text{eval}})|\mathbf{X}_s^{\text{eval}}) \ . \qquad (6)$$

By optimizing Objective (**6**), we encourage the kernel hyperparameters $\theta$ to capture the implicit prior distribution over functions induced by the models in the family $\mathfrak{F}$ as applied to the datasets in the family $\mathfrak{D}$. Algorithm (**1**) gives the complete surrogate model training and evaluation process.

## 3.2 Why use (GP) surrogate models for NNs?

By estimating a prior over functions for a neural network family directly from neural network behavior, we aim to capture shared properties that determine the model family's behavior on data, *i.e.,* **the model family's inductive biases**. There is strong evidence that the inductive biases of neural networks (*e.g.,* invariances and equivariances, Markovian assumptions, compositionality) and not just data, play an important role in their performance (Poggio, Mhaskar, et al. 2017; Tiňo et al. 2004; Lin and Tegmark 2017; Fukushima 2004; Werbos 1988). Moreover, deep NNs are highly overparametrized models that can nevertheless generalize well, prompting interest in implicit regularization mechanisms that bias NNs towards learning simpler solutions (Soudry et al. 2018; Poggio, Kawaguchi, et al. 2018; Neyshabur et al. 2017). More broadly, the extrapolation behavior of any learning machine is underdetermined by data alone and therefore depends on its inductive biases (Mitchell 1980).

GPs, in particular, offer several advantages as surrogate models of NNs. Firstly, GPs **are flexible models that are also often interpretable** in the sense that the learned hyperparameters can provide insights into properties of the datasets on which they are trained (Wilson and Adams 2013). As an example, many covariance functions have separate lengthscales for each input dimension. An inverse lengthscale captures an input dimension's "importance;" in Section (**4.4**), we demonstrate that we can use these lengthscales

---

[2]We consider both untrained and trained neural networks, where an untrained network is a special case of a trained network with the number of training iterations at 0; we thus describe the framework only for trained networks.

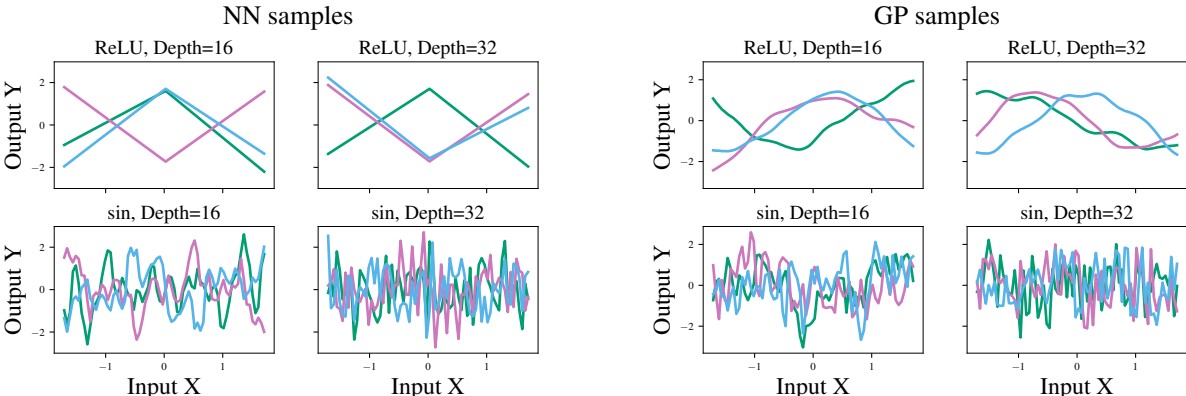

**Figure 2: Demonstration: Comparing learned GP priors with NN priors.** Samples from GP prior (**right**) with kernel hyperparameters inferred from the predictions of NN families (**left**). GPs are flexible enough to capture properties of each NN family; for example, the samples from the learned GP prior reflect the quickly varying behavior of the 32-layer sinusoidal NNs and the increasing-decreasing behavior of rectifier NNs.

to predict generalization behavior, suggesting that the GP surrogate representation is practically useful in automating model selection.

Secondly, the use of GP surrogate models is also motivated by the **theoretical connections between GPs and NNs.** Neal (1996) showed that a prior over the parameters of certain single-layer multi-layer perceptrons (MLPs) converges to a GP as the MLP's width approaches infinity, and recent works (Lee et al. 2017; Matthews et al. 2018; Novak, Xiao, Bahri, et al. 2019; Garriga-Alonso et al. 2019; Yang 2019) have extended this correspondence to deep MLPs and more modern NN architectures. Connections between GPs and NNs can provide insight because they transform the priors implicit in NNs designs into explicit priors expressed through a GP. However, our strategy to derive such a connection differs from this prior theoretical work that derives analytic kernels for limiting cases of NNs—we take an empirical approach by learning GP kernels directly from the predictions of arbitrary classes of finite NNs.

Lastly, **GPs have a tractable marginal likelihood.** Probabilistic models allow us to express inductive biases in the form of an explicit prior distribution, but the marginal likelihood is intractable for most complicated Bayesian models. In contrast, for GPs, the marginal likelihood has a closed form expression, which means that we can optimize it directly instead of resorting to approximations.

### 3.3   Demonstration: Comparing learned GP priors with NN priors

We briefly demonstrate the surrogate modeling framework of Section (**3.1**). As a simple sanity check, we verify that GP surrogates learned from varying NN families exhibit meaningful variation in behavior. To do this, we learn GP priors from varying NN families and compare the learned priors with the NN families.

**NN hyperparameters.**   We consider ensembles of 50 randomly initialized NNs with rectified linear unit (ReLU) or sine (sin) activations and 16 or 32 hidden layers of 128 hidden units each. We randomly initialize the weights about zero with weight variance $\sigma_w^2 = 1.5$ and bias variance $\sigma_b^2 = 0.05$.

**GP surrogate.**   For each ensemble, we learn the hyperparameters of a randomly initialized SMK with $Q = 10$ mixture components by optimizing Objective (**6**) for 350 iterations with batch gradient descent and the adaptive momentum (Adam) optimizer (Kingma and Ba 2015) with a learning rate $\eta = 0.1$. We choose the kernel hyperparameters with the highest objective value across three random initializations.

**Results.**   We plot NN predictions and samples from the learned GP priors in Fig. (**2**). The learned GP captures the periodicity of the sinusoidal neural networks (sinusoidal NNs), and partially captures the increasing-decreasing behavior of rectifier neural networks (rectifier NNs) about a cusp; though, due to the

SMK parameterization, it cannot capture the discontinuity at the cusp. The GP also captures differences in depths for the sinusoidal NNs: The GP prior samples for the 32-layer networks are quickly varying, indicating shorter lengthscales have been learned. Taken together, the results of this demonstration show that GP surrogates can capture certain NN behavior.

## 4 Experiments

We provide a series of demonstrations of the value of the approach of Section (**3**). Each experiment aims to investigate the properties of one or more neural network families, specified by **neural network (NN) hyper-parameters**, as evaluated on one or more dataset families, parameterized as **target functions**, by analyzing the corresponding **Gaussian process (GP) surrogate model**. In Sections (**4.1**) and (**4.2**), we capture previously established NN phenomena, while in Sections (**4.3**) and (**4.4**), we predict NN generalization behavior.

### 4.1 Reproduction: Capturing spectral bias in NNs

Rahaman et al. (2019) demonstrated that deep rectifier NNs exhibit *spectral bias*, the preference to learn lower frequencies in the target function before higher frequencies. To demonstrate this, the authors studied the Fourier spectrum of rectifier NNs fit to a sum of sinusoidal functions of varying frequencies. In this section, we take an alternative approach: We learn kernels from NN predictions at various stages of training and demonstrate that the evolution of these learned kernels captures the spectral bias.

**NN hyperparameters.** As in Rahaman et al. (2019), we train an NN with 6 hidden layers of 256 units and ReLU activations using full-batch gradient descent with Adam and a learning rate of $\eta = 3 \times 10^{-4}$.

**Target function.** The target functions are sums of sine functions with frequencies in $(5, 10, \ldots, 45, 50)$ and phases drawn from $U(0, 2\pi)$, evaluated at 200 points evenly spaced between $[0, 1]$, as in Rahaman et al. (2019).

**GP surrogate.** We learn the parameters of a spectral mixture kernel (SMK) with $Q = 10$ mixture components by optimizing Objective (**6**) with Adam for 350 iterations with a learning rate of $\eta = 0.1$. Since the marginal likelihood of the SMK is multi-modal in its frequency parameters, we repeat this optimization for three different random initializations of the kernel parameters and choose the hyperparameters with the largest marginal likelihood value (the value of Objective (**6**)). We randomly initialize the length-scales $v_i$ by sampling from a truncated normal distribution whose variance depends on the maximum distance between input points. We set the signal variances $w$ to the variance of the target function values divided by the number of mixture components. The frequency hyperparameters of the SMK are sometimes initialized by sampling from a uniform distribution whose upper limit is the Nyquist frequency (Wilson and Adams 2013); since this target function's largest frequency is smaller than the Nyquist frequency, we instead set a smaller frequency as the upper limit.

**Results.** Fig. (**3**) displays the NN predictions and the kernel of the corresponding GP surrogate at different iterations of NN training. The kernel function, which is given in Eq. (5), reflects how the similarity between function values varies with the distance between their input points.[3] The structure of the learned kernel reflects the properties of the NN family: Initially, the learned kernel only captures low frequencies in the NN's predictions—reflected in the long period of the kernel—consistent with the spectral bias of Rahaman et al. (2019). However, as training progresses, the periodicity of the learned kernel reflects both low and high frequencies.

### 4.2 Reproduction: Depth pathologies in randomly initialized NNs

Hyperparameter selection in NNs is not always theoretically grounded. Many recent studies thus characterize how different hyperparameter choices (*e.g.,* depth, width) affect the properties of NNs at random initialization (Schoenholz et al. 2017; Yang 2019; Xiao et al. 2018). Towards that end, recent work showed that

---

[3]Since the SMK is a stationary covariance function, we graph against the distance between input points rather than the absolute value of the input points themselves.

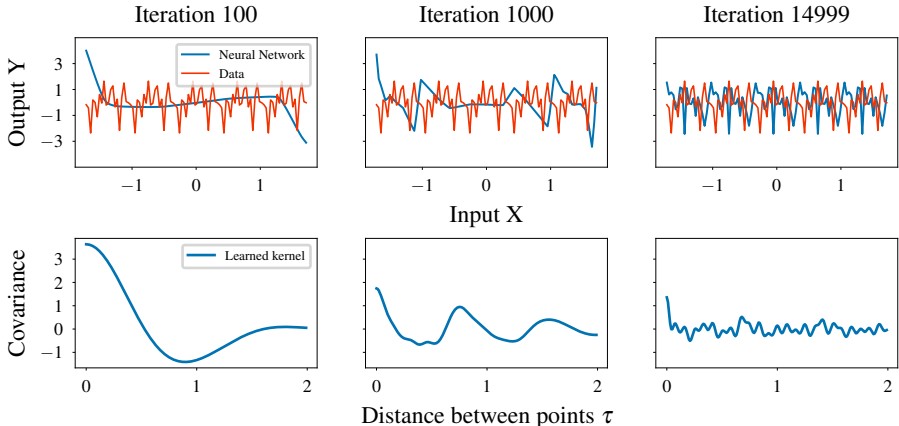

**Figure 3: Capturing spectral bias in neural networks.** (**Top**) Neural network predictions as training progresses on the sum-of-sines target function described in Section (**4.1**). (**Bottom**) Spectral mixture kernel fit to neural network predictions as training progresses. The kernel reveals a spectral bias for this neural network family, with the range of spectral frequencies expressed in the kernel increasing with the number of iterations of training.

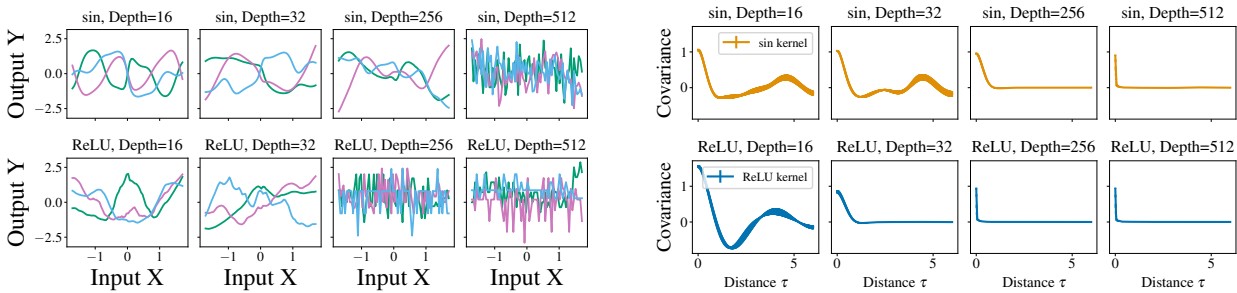

**Figure 4: Depth pathologies in randomly initialized neural networks.** Predictions of neural networks (**left**) from neural network families of different activations (**rows**) and varying depths (**columns**); mean and standard error of the covariance of the corresponding surrogate model kernels (**right**). The covariance is aggregated across 10 kernels learned from 10 different 50-member neural network ensembles from a given family. Greater depth results in kernels with shorter lengthscales, with this pathology emerging earlier in rectifier NNs; this result is consistent with prior work on pathologies of deep neural networks.

increasing depth could actually induce pathologies in randomly initialized NNs (Labatie 2019; Duvenaud, Rippel, et al. 2014). For example, Duvenaud, Rippel, et al. (2014) proved that increasing depth in a certain class of infinitely wide NNs produces functions with ill-behaved derivatives. As a result, these functions are quickly varying in the input space.

We empirically study a similar pathology—quick variation in input space—that emerges in randomly initialized, finite-width, finite-depth NNs. To do this, we fit GP surrogates to randomly initialized NN ensembles of varying depths and activation functions and inspect how the learned kernels change with depth. If NNs exhibit this pathology, the learned covariance will decay sharply with distance.

**NN hyperparameters.** We consider families of NNs of varying activation functions (sin ($a \sin(bx + c)$) and ReLU ($\max(0, x)$)) and varying depths (from 16 to 512 layers). From each family, we sample an ensemble of 50 randomly initialized NNs, each with 128 hidden units in each layer. We randomly initialize NN weights about zero with weight variance $\sigma_w^2 = 1.5$ and bias variance $\sigma_b^2 = 0.05$.

**GP surrogate.** We sample 10 ensembles of 50 randomly initialized NNs, and learn an SMK kernel by optimizing Objective (**6**) separately for each ensemble, running Adam (Kingma and Ba 2015) for 750 iterations with a learning rate of $\eta = 0.1$. We choose the kernel hyperparameters with the highest mean marginal likelihood among three random initializations. To ensure our results are robust across random ensembles, we

consider an averaged learned kernel: Suppose we have $n$ kernels, $k_1(\cdot), \ldots, k_n(\cdot)$, learned from $n$ different ensembles from the same family. The average learned kernel, $\bar{k}$, is defined as $\bar{k}(\tau) = \frac{1}{n} \sum_{i=1}^{N} k_i(\tau)$.

**Results.**  Fig. (4) plots the average learned kernels for NN families with varying activation functions and depths, as well as the predictions of those NN families. Across both activation functions, the learned kernels reveal a pathology: For large depths, the covariance (Fig. (4), right) sharply decays towards zero with distance. The NN predictions (Fig. (4), left) explain this property of the learned kernels: At large depths, the deep NNs vary quickly in the input domain, which causes the SMK to learn short lengthscales. Interestingly, this pathology emerges at different depths for different activation functions: We see rectifier NNs exhibit this pathology with 256 layers while sinusoidal NNs exhibit this pathology with 512 layers.

### 4.3   Ranking NN generalization with the GP marginal likelihood

In previous sections, we demonstrated that GP surrogate models could yield insight into NN behavior. The benefits of GPs extend beyond this. Since the GP marginal likelihood has a closed form expression, many have advocated for using the marginal likelihood in model selection and as an indicator of expected generalization performance (Mackay 1992). In this section, we leverage the learned GP surrogate to *rank NNs by their generalization error with the GP marginal likelihood.* In particular, we learn GP surrogates from different NNs at random initialization, and we then study if the marginal likelihood of the surrogates can rank the NNs by test error after training. In the following experiments with varying classes of NN families, we find that we can indeed predict test error using the marginal likelihood of the training set under the learned surrogate GP.

#### 4.3.1   The idealized case: Large-width NNs

Before we consider arbitrary NN families, we check that the marginal likelihood is predictive in an idealized setting. In particular, we consider large-width NNs whose infinite-width analogs are equivalent to GPs (Lee et al. 2017). If the marginal likelihood is not predictive in this case in which the kernel function can be analytically determined, it is unlikely to be useful in a general setting where the kernel is learned and GPs approximate NNs priors but are not equivalent.

**NN hyperparameters.**  We consider NNs with sin or Gauss error function (erf)[4] activations and 2 hidden layers of 1024 units each. We randomly initialize the weights about zero with weight variance $\sigma_w^2 = 1.5$ and bias variance $\sigma_b^2 = 0.05$. We train an ensemble of 50 randomly initialized NNs from each family using full-batch (vanilla) gradient descent with learning rates of $\eta \in \{0.01, 0.1\}$.

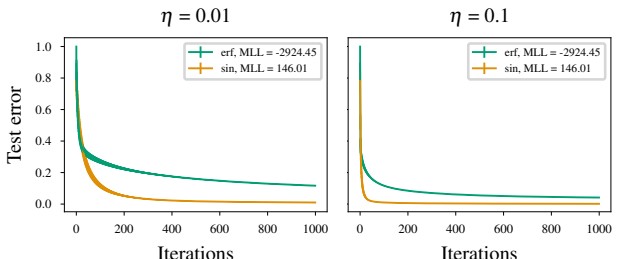

**Figure 5: Ranking generalization from MLL in large-width NNs.** Mean and standard error of the test MSE of large-width sinusoidal and erf NNs trained with learning rates $\eta = 0.01$ **(left)** and $\eta = 0.1$ **(right)** on the target function of Section (4.3.1). The MLL of the target function under the surrogate model corresponding to the limiting kernel for each model family is shown in the legend. Consistent with expectations, the model family whose surrogate assigns higher MLL to the target function achieves lower test error for both values of $\eta$.

**Target function.** The target function is $\sin(0.5x)$.

**GP surrogate.**  We do not learn a kernel from NN predictions as in previous sections. Instead, we use the kernels corresponding to the infinite width analogs of the NNs using the neural-tangents package (Novak, Xiao, Hron, et al. 2020).

**Results.**  Fig. (5) compares the performance of these NN families along with the marginal likelihood of the target function under the surrogate model. The performance (mean-squared error (MSE) on the test set) is averaged across each ensemble of NNs. The marginal log-likelihood (MLL) of the target function is higher for the better-performing NN family.

---

[4]Here, erf is defined as $a\,\mathrm{erf}(bx) + c$, where $\mathrm{erf}(x) = \frac{2}{\sqrt{\pi}} \int_0^x e^{-t^2}\,dt$.

### 4.3.2  Small width neural networks and learning the kernel

In the previous experiment, we showed that the marginal likelihood could be predictive when we consider large-width NNs and when we use a corresponding, analytically derived kernel. Is the marginal likelihood predictive when we consider smaller-width NNs and when we learn the kernel empirically?

**NN hyperparameters.**  We consider ensembles of width 16, depth 4 NNs from two families: NNs with sin activations and NNs with ReLU activations. We randomly initialize weights about zero with weight variance $\sigma_w^2 = 1.5$ and bias variance $\sigma_b^2 = 0.05$. We train an ensemble of 50 randomly initialized NNs from each family on the target functions using full-batch gradient descent with a learning rate of $\eta = 0.1$.

**Target function.**  The target function families mirror the NN model families: We collect predictions from randomly initialized, width 16, depth 4 NNs with sin or ReLU activations. These target functions are a useful sanity check, as the inductive biases of the model families are perfectly suited for a target function family.

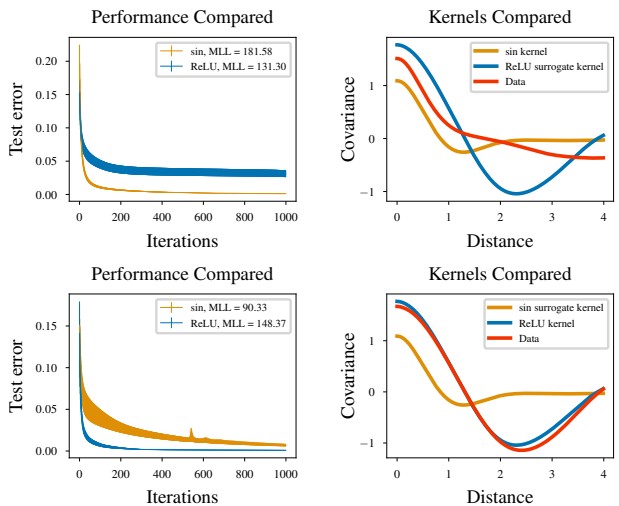

**GP surrogate.**  For each ensemble, we learn the hyperparameters of an SMK with $Q = 5$ mixture components by optimizing Objective (**6**) across the ensemble. To optimize, we randomly initialize the kernel hyperparameters and run Adam for 250 iterations with a learning rate of $\eta = 0.1$. We initialize the frequency parameters by sampling from a uniform distribution whose upper limit is the Nyquist frequency. We choose the kernel hyperparameters with the highest objective value across three random initializations.

**Results.**  In Fig. (**6**), we compare the performances of the two NN families on the two target function families. We also display the kernels learned from NN behavior (*sin surrogate kernel* or *ReLU surrogate kernel*) and learned from the target function family (*data kernel*) directly. Across both experiments, the MLL averaged across the target function family of the better-performing NN family is higher. In general, the structure of a learned kernel reflects the properties of the learned GP prior, and so we can compare kernels to assess similarity between target function and NN families. We see that the data kernel provides a better qualitative match to the kernel of the better-performing model family.

**Figure 6: Ranking generalization from MLL in small-width NNs.** Mean and standard error of test MSE **(left)** of small-width sinusoidal and rectifier NN ensembles on sin **(top)** and ReLU **(bottom)** target function families, with the target function MLL under the surrogate learned from each model family in the legend. Covariance **(right)** of surrogate kernels alongside data kernels learned from the sin **(top)** and ReLU **(bottom)** target function families. Even in the small-width regime and when the kernel is learned, the model family whose surrogate assigns a higher MLL to the target function attains lower error **(left)**; the surrogate kernel learned from the better-performing model family better matches the data kernel **(right)**.

### 4.3.3  Systematic study of various learning rates and architectures

In this last experiment on ranking generalization performance, we establish that Gaussian process surrogates reliably rank performance across a range of learning rates and gradient descent algorithms.

**NN hyperparameters.**  We consider ensembles of randomly initialized NNs with sin or ReLU activations and 1 or 3 hidden layers with 256 hidden units in each layer. We randomly initialize the weights about zero with weight variance $\sigma_w^2 = 1.5$ and bias variance $\sigma_b^2 = 0.05$. We train 50 randomly initialized NNs from each family using either vanilla full-batch gradient descent with a constant learning rate of $\eta = 0.01$, or Adam (Kingma and Ba 2015) using learning rates of $\eta \in \{0.0003, 0.003\}$.

**Target function.**  We consider a target function of $\sin(0.5x)$.

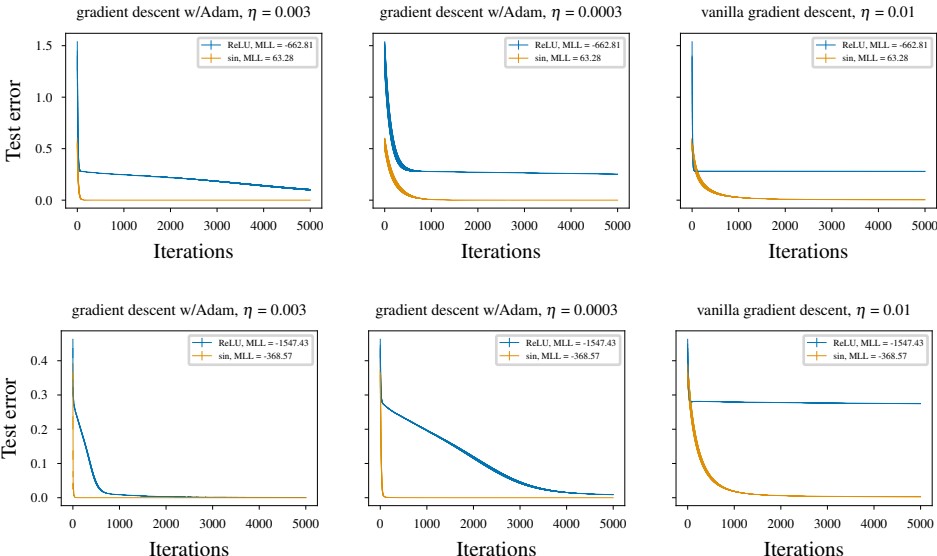

**Figure 7: Ranking generalization performance from MLL across different learning algorithms and architectures.** Each panel displays mean and standard error of test MSE of an NN family trained on the target function $\sin(0.5x)$ with noise; legend displays MLL of the training data under the surrogate for one of two NN families: 1-layer (256 hidden units) sinusoidal or rectifier NNs **(top)**; 3-layer (256 hidden units) sinusoidal or rectifier NNs **(bottom)**. NNs are trained with batch gradient descent with Adam (learning rates $\eta = 0.003$, $\eta = 0.0003$) or vanilla batch gradient descent ($\eta = 0.01$). Across architectures and learning algorithms, the NN family whose surrogate assigns higher MLL to the target function achieves lower test error.

**GP surrogate.** For each ensemble, we learn the hyperparameters of an SMK with $Q = 5$ mixture components by optimizing Objective (6) across the ensemble. To optimize, we randomly initialize the kernel hyperparameters and run Adam for 250 iterations with a learning rate of $\eta = 0.1$. We choose the kernel hyperparameters with the highest objective value across three random initializations. To randomly initialize the frequency parameters, we uniformly sample from the real-valued interval $(0, 25]$.

**Results.** In Fig. (7), we find that the marginal likelihood of the better-performing NN family is higher. The marginal likelihood depends on the diagonal noise $\sigma_n^2$ added to the Gram matrix (Eq. (3)). We find that our result are robust across three levels of this diagonal noise $(10^{-3}, 10^{-4}, 10^{-5})$. These results suggest we can rank these NN families when they are not in the asymptotic regime and when we learn the kernel, in contrast to Section (4.3.1), as well as when *a priori* no model family should perform better, unlike Section (4.3.2).

## 4.4 Predicting the NN generalization gap with the GP marginal likelihood

In the previous section, we predicted generalization using kernels learned from randomly initialized NNs. However, some design choices do not affect NN properties at random initialization but may still strongly influence generalization (*e.g.,* learning algorithm). Motivated by this, we characterize trained NN properties on the *validation set* and compare these properties to the training data. We focus on the validation set because it is more informative of extrapolation. If the NN extrapolates well, its predictions on the validation set should be "similar" in some sense to the dataset. On the other hand, significant discrepancies could indicate poor extrapolation. This intuition motivates our analysis.

In particular, we learn a kernel from the training data and a kernel from NN predictions on a validation set. We then quantitatively compare these kernels by computing a metric we describe in more detail later. We find that a lower similarity between these kernels correlates with a larger *generalization gap* (*i.e.,* poorer extrapolation), defined as the difference between test error and training error (*e.g.,* Jiang et al. 2020).

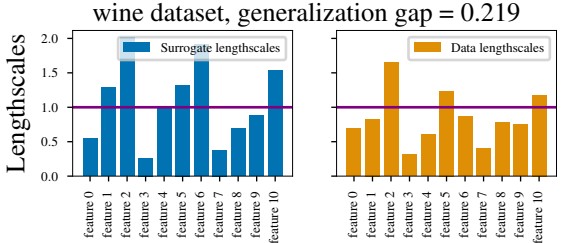 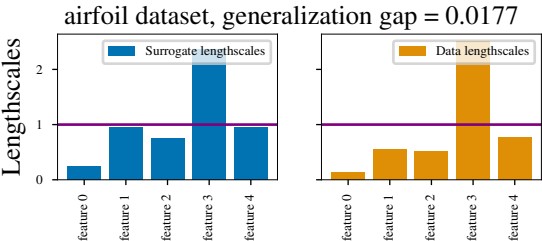

**Figure 8: Qualitative connection between lengthscale profile discrepancy and generalization gap.** Each subfigure compares normalized lengthscales learned from neural network predictions on validation set (*i.e.,* surrogate lengthscales) after training and normalized lengthscales learned from training data (*i.e.,* data lengthscales). A lengthscale greater than 1 indicates an "unimportant" feature. The title indicates the UCI dataset and generalization gap defined in Fig. (**9**). Data and surrogate lengthscales for some features are different (*e.g.,* features 1, 4, 6), reflected in a high generalization gap (**left**). Data and surrogate lengthscales for the same features are generally similar, reflected in a low generalization gap (**right**). This suggests a connection between the generalization gap and discrepancy between surrogate and data lengthscales.

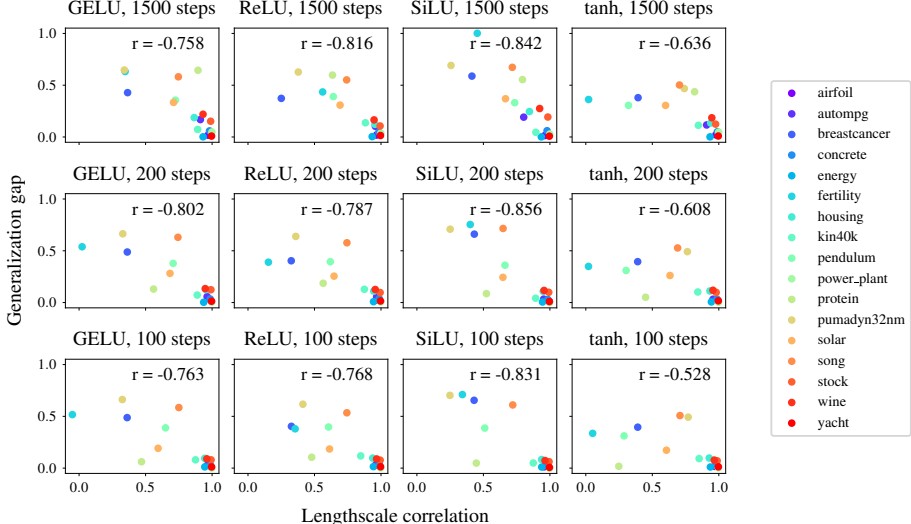

**Figure 9: Inverse relationship between generalization error and lengthscale correlation on UCI datasets.** Each point represents the lengthscale correlation (between surrogate and data lengthscales) and the generalization gap for a neural network ensemble to which the surrogate model is fit, on a single UCI dataset. Each panel corresponds to a particular neural family; see Section (**4.4**) for details about hyperparameters of these families, including architectures. Colors correspond to a particular UCI dataset. Across datasets and architectures, a larger lengthscale correlation (*i.e.,* higher similarity between the data and surrogate representations) corresponds to a lower generalization gap (*i.e.,* better extrapolation).

**NN hyperparameters.** We train ensembles of randomly initialized NNs with sigmoid-weighted linear unit (SiLU) (Elfwing et al. 2018), Gaussian error linear unit (GELU) (Hendrycks and Gimpel 2016), ReLU (Fukushima 1975; Nair and Hinton 2010), or hyperbolic tangent (tanh) activations, and two layers of 128 hidden units. We use the LeCun normal initialization with a scale of 1.5 (LeCun et al. 2012). We train 25 NNs with full-batch gradient descent using Adam with a learning rate of $\eta = 0.003$. We want to assess if our approach can distinguish between NNs with similar training behavior but varying generalization performance. We train NNs either for a maximum number of iterations, a hyperparameter, or until training error reaches zero.

**Target functions.** We consider a set of naturalistic regression tasks from the UC Irvine Machine Learning Repository (UCI) dataset (Dua and Graff 2017), spanning a range of dataset sizes and input dimensions. We split each of the datasets into a 72/8/20 train/validation/test split. Both the data input and output are standardized by mean-centering and dividing by the standard deviation dimension-wise so that the target

values and each dimension of the data input have near zero mean and unit variance. We subsample 2,000 datapoints for datasets with more than 2,000 datapoints, as in Simpson et al. (2021) and Liu et al. (2020).

**GP surrogate.** We learn a *data kernel* directly from the training dataset. We also learn a *surrogate kernel* from NN predictions on the validation set. In both cases, we use the Matérn kernel (MK) since the SMK can struggle for higher-dimensional inputs. We learn a separate lengthscale for each input dimension (*i.e.,* feature) of the data. We denote the lengthscales for a kernel as its *lengthscale profile.* We call the data kernel's lengthscales the *data lengthscales* and the surrogate kernel's lengthscales the *surrogate lengthscales.* To quantify the mismatch between NN validation predictions and the training data, we consider the *correlation in lengthscale profiles across features.* This is the correlation between the data and surrogate lengthscales.

**Results.** Fig. (**8**) gives intuition for our more general result in Fig. (**9**). For two UCI datasets, we compare the data lengthscales and the surrogate lengthscales for a two-layer GELU NN. The vertical axis corresponds to (normalized) learned lengthscales for each input dimension.[5] When the generalization gap is small, the data kernel and surrogate kernel are similar; the same features have similar lengthscales (Fig. (**8**), right). When the generalization gap is large, the data kernel and surrogate kernel have discrepancies. For example, the surrogate lengthscales for features 1 and 6 are larger than 1, but the data lengthscales for feature 1 and 6 are smaller than 1 (Fig. (**8**), left).

In Fig. (**9**), we summarize our results across different architectures, datasets, and maximum training iterations. We display the generalization gap against the correlation in lengthscale profiles across features. The similarity in lengthscale profiles negatively correlates with generalization gap across a range of architectures and max iterations. The Pearson correlation coefficients range from $-0.856$ to $-0.528$. In Appendix (**B**), we additionally demonstrate that these results are insensitive to outlier datasets by performing a dataset-sensitivity analysis.

### 4.5 Correlating NN generalization with the GP marginal likelihood and lengthscales

In this section, we extend the analysis in the previous section to a larger hyperparameter sweep of models.

**NN hyperparameters.** We train ensembles of randomly initialized NNs with four activations (SiLU (Elfwing et al. 2018), GELU (Hendrycks and Gimpel 2016), ReLU (Fukushima 1975; Nair and Hinton 2010), or tanh), two different depths (2 and 4), two different numbers of hidden units (32, 64) in each layer, and two sets of learning rates with Adam (0.03, 0.003); this leads to a total of 32 different combinations of hyperparameters. We use the LeCun normal initialization with a scale of 1.5 (LeCun et al. 2012). We train 50 NNs with full-batch gradient descent using Adam with two varying learning rates. We train NNs for 500 iterations.

**Target functions.** We consider a set of naturalistic regression tasks from the UCI dataset (Dua and Graff 2017), spanning a range of dataset sizes and input dimensions. We consider a set of UCI datasets that satisfy two criteria: the number of unique feature values for each dimension is greater than 2 and the range in lengthscale correlations between neural network and datasets is greater than 0.025. The first condition ensures that we can learn sensible lengthscales for the dataset. The second condition ensures there is meaningful variation in neural network behavior within a single dataset.

We split each of the datasets into a 72/8/20 train/validation/test split. Both the data input and output are standardized by mean-centering and dividing by the standard deviation dimension-wise so that the target values and each dimension of the data input have near zero mean and unit variance. We subsample 2,000 datapoints for datasets with more than 2,000 datapoints, as in Simpson et al. (2021) and Liu et al. (2020).

**GP surrogate.** We learn a *data kernel* directly from the training dataset. We also learn a *surrogate kernel* from NN predictions on the validation set. In both cases, we use the Matérn kernel (MK) since the SMK can

---

[5]For this visualization, we divide the learned lengthscale for each dimension by the difference between the maximum feature value and minimum feature value for each dimension. By doing so, we can interpret a lengthscale that is much greater than 1 as suggesting that the NN predictions do not vary much along that dimension.

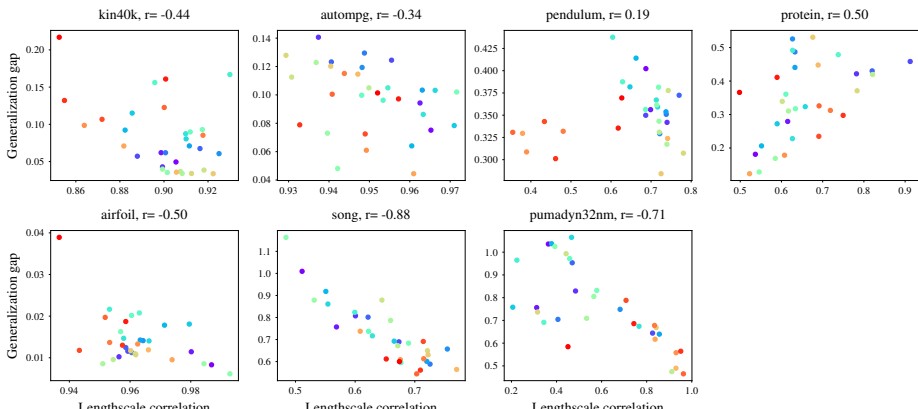

**Figure 10: Inverse relationship between generalization error and lengthscale correlation on UCI datasets across hyperparameter sweep of neural networks.** Each point represents the lengthscale correlation (between surrogate and data lengthscales) and the generalization gap for a neural network ensemble to which the surrogate model is fit. Each panel corresponds to a particular UCI dataset; see Section (**4.5**) for details about hyperparameters of these families, including architectures. In 5/7 datasets, a larger lengthscale correlation (*i.e.,* higher similarity between the data and surrogate representations) corresponds to a lower generalization gap (*i.e.,* better extrapolation).

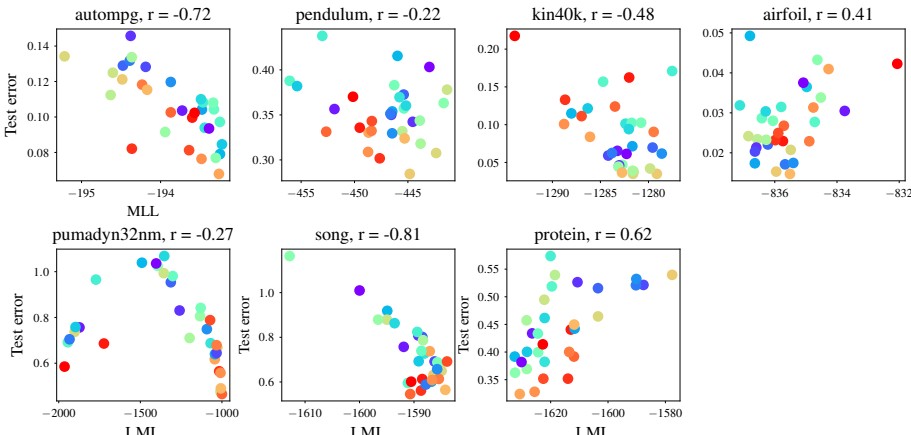

**Figure 11: Inverse relationship between test error and marginal likelihood on UCI datasets** Each point represents the marginal likelihood of the training data (using the kernel learned for a neural network family) and the test error for a neural network ensemble to which the surrogate model is fit. Each panel corresponds to a particular UCI dataset; see Section (**4.5**) for details about hyperparameters of these families, including architectures. Each color corresponds to a particular neural network ensemble. In 5/7 datasets, a larger marginal likelihood correlates with lower test error.

struggle for higher-dimensional inputs. We learn a separate lengthscale for each input dimension (*i.e.,* feature) of the data. We denote the lengthscales for a kernel as its *lengthscale profile*. We call the data kernel's lengthscales the *data lengthscales* and the surrogate kernel's lengthscales the *surrogate lengthscales*. To quantify the mismatch between NN validation predictions and the training data, we consider the *correlation in lengthscale profiles across features*. We also consider the marginal likelihood of the training dataset under the learned kernel (*i.e.,* the Matern Kernel with the surrogate lengthscales) for each family of neural networks.

**Results.**   In Figure 10, we plot the generalization gap against the correlation in lengthscale profiles across features. In contrast to Figure 9, each panel corresponds to a particular dataset and each data point corresponds to a particular neural network family with a set of hyperparameters described above.

The majority of datasets exhibit negative correlation between correlation the in lengthscale profiles and generalization gap with two exceptions (pendulum, protein). We explain these exceptions. For the pendulum

dataset, the positive correlation is driven by neural networks with tanh activations (the dots in different shades of red). We think the Matern kernel may struggle to model the tanh networks. This is consistent with Figure 9 where the correlations were consistently lower for the tanh networks. For the protein dataset, the positive correlations might be related to challenges in fitting GPs to this dataset; recent work showed that exact GP regression on a training dataset (without any subsampling) from the protein dataset attains high test RMSE (K. A. Wang et al. 2019).

In Figure 11, we plot the test error against the marginal likelihood of the training data under the various learned kernels for each neural network ensemble. That is, each point corresponds to the marginal likelihood of the training data using the kernel learned from each neural network family. We do this for several datasets, indicated by the figure subtitle. The marginal likelihood correlates with test error on several of the datasets. Consistent with previous work (Lotfi et al. 2022), we find that this result is sensitive to the jitter (which we set to 0.5).

## 5    Discussion

In this paper, we illustrated the potential of modeling neural networks with Gaussian process surrogates. We empirically characterized phenomena in neural networks by interpreting kernels learned directly from neural network predictions, capturing the spectral bias of deep rectifier networks (Section (**4.1**)) and pathological behavior in deep, randomly initialized neural networks (Section (**4.2**)). We further demonstrated that Gaussian process surrogates could predict neural network generalization by ranking test error performance by marginal (Section (**4.3**)) and by quantifying the generalization gap via a surrogate-data kernel discrepancy (Section (**4.4**)). Taken together, these results suggest that Gaussian process surrogates may be a valuable empirical tool for investigating deep learning, and future work could aim to use this framework to complement existing approaches to interpretability (*e.g.,* Ribeiro et al. 2016) and extrapolation (*e.g.,* Xu et al. 2021).

We note a couple of limitations of our current study. First, though the framework is in principle applicable to broader settings, we restricted this first exploration to regression tasks and feed-forward neural network architectures. A broader study of more architectures on more types of tasks would be challenging due to the need to scale Gaussian processes but potentially rewarding, as characterizing properties of neural networks as used in practice is a significant open problem with far-reaching implications (Sejnowski 2020). Second, we learn point estimates of kernel hyperparameters (type II maximum likelihood; Gelman et al. 2013). Although this is standard, we could infer the posterior over hyperparameters using Markov chain Monte Carlo (MCMC) or variational inference (Lalchand and Carl Edward Rasmussen 2020; Murray and Adams 2010; Simpson et al. 2021) to perform a fully Bayesian analysis. We also could explore a richer set of kernels, such as compositional kernels (Duvenaud, Lloyd, et al. 2013). These directions are exciting avenues for future work.

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

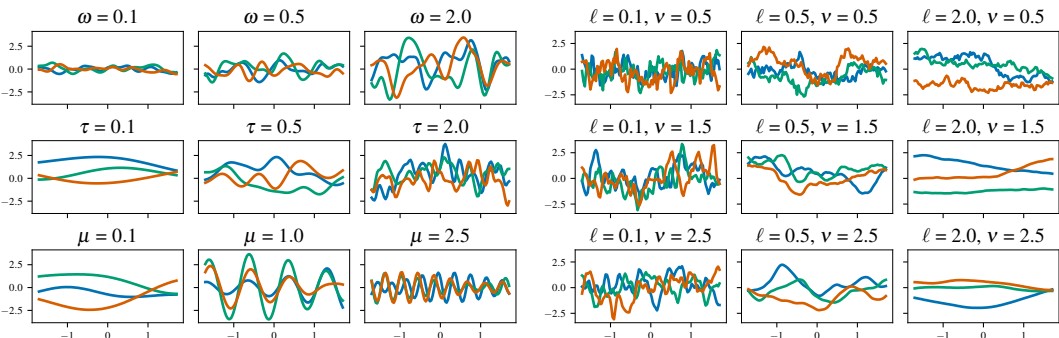

**Figure 12: Illustrating the effect of GP kernel hyperparameters on the GP prior.** (**Left**) Samples from a GP prior with SMK with varying mixture weights $\omega$, mixture scale $\tau$, and mixture means $\mu$. (**Right**) Samples from a GP prior with Matern kernel with varying $\nu$ and $\ell$ (lengthscale). GPs are flexible models whose properties can be controlled through hyperparameters.

# A  Properties of the spectral mixture kernel and the Matérn kernel

We describe how the various hyperparameters of the SMK and MK kernel affect the GP prior. We begin with the spectral mixture kernel. The mixture weights $w$ are signal variances and control the scale of the function values. The mixture means ($\mu$) encode periodic behavior. The variances ($\tau$) are (inverse) lengthscales, which control the smoothness. The (ARD) MK kernel has lengthscales $\theta$, which controls the smoothness of the function with respect to each dimension. $\nu$ is another hyperparameter that also modulates smoothness, and the Matern covariance function admits a simple expression when $\nu$ is a half-integer. $\nu = 2.5$ corresponds to twice differentiable functions and $\nu = 1.5$ corresponds to once differentiable functions.

In Fig. (**12**), we vary the hyperparameters of the SMK ($w, \mu, \tau$)and Matern kernels ($\nu, \theta$) and illustrate how they impact the prior over functions.

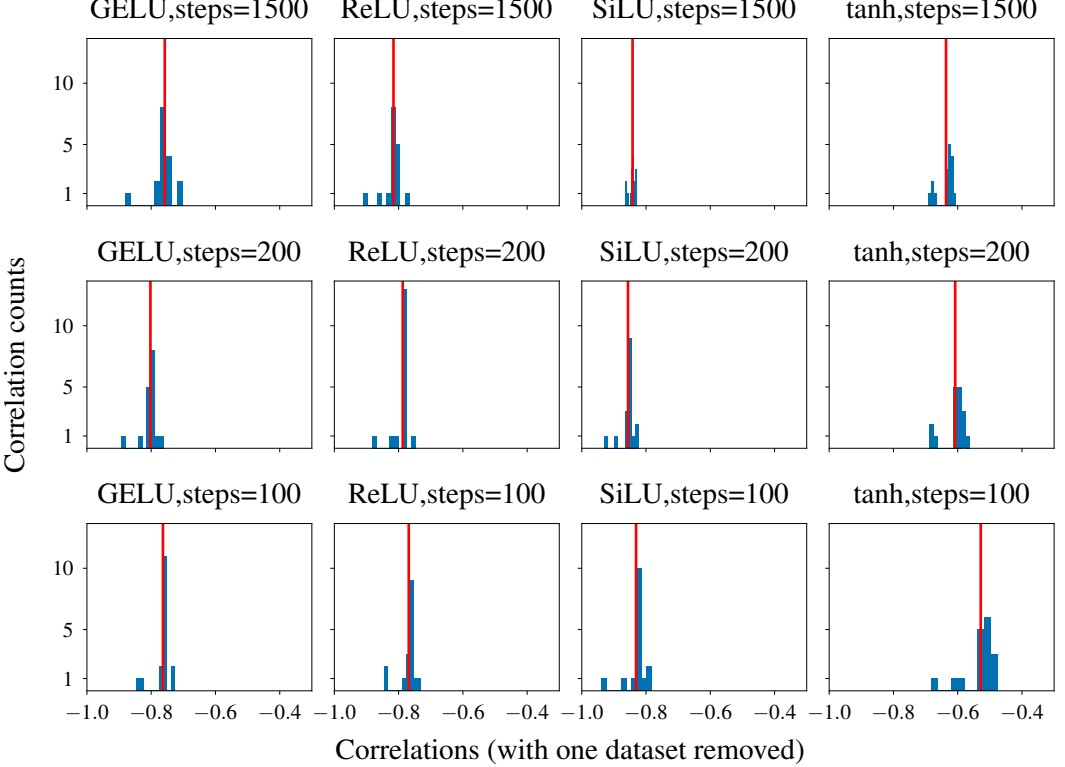

**Figure 13: Sensitivity analysis of generalization gap and lengthscale profile relationship.** Each panel a histogram and mean (red line) of correlations obtained by recomputing the correlation between lengthscale profile correlation and generalization gap after removing each UCI dataset. Across datasets and architectures, even when a single dataset is removed, there remains an negative correlation between generalization gap and lenthscale profile correlation. Therefore, the inverse relationship between generalization gap and lengthscale profile correlation demonstrated in Fig. (**9**) is robust to outlier datasets.

## B  Correlation sensitivity

We present some additional results to supplement our analysis from Section (**4.4**) where we demonstrated that discrepancy in lengthscale profiles between data and neural network predicts the generalization gap. Correlation can be sensitive to outliers. Does any single dataset account for the negative correlations? To answer this, we characterize how the correlation changes as a result of dropping each dataset. Specifically, for each UCI dataset, we remove that dataset and then compute the correlation between lengthscale profile correlation and generalization gap for the remaining datasets. We plot the resulting distribution of correlations in Fig. (**13**). We find there is a tight spread around the correlation computed from all the UCI datasets. Importantly, when we remove any UCI dataset, we still see moderate to high negative correlations between lengthscale profile correlation and generalization gap.

