# OpenReview forum: "Gaussian process surrogate models for neural networks"
_TMLR — Rejected by TMLR_

### Review · Reviewer_iVRX · 2022-08-29

**Summary Of Contributions:**

The authors propose to model a (un)trained neural network (NN) using Gaussian processes (GPs) with various kernels.

After obtaining a neural network's outputs on a set of evaluation sets (with a network trained separately on a respective training set), a parametric GP is fit to maximize its marginal likelihood of these outputs.

Then, the GP kernel can be analyzed to infer certain properties of the respective NN, such as predicting its generalization, model selection (against other NNs), identifying pathologies, or describing training dynamics.


## Are the claims made in the submission supported by accurate, convincing and clear evidence?

The evidence is clear and accurate, but not very convincing in my opinion. As I detail in **"Weaknesses"**, the paper doesn't demonstrate a setting where a particular insight is derived about a NN from a surrogate GP, which would be harder to see in the NN directly. The evidence for the claim about GP marginal likelihood correlating with NN generalization (which could be such an insight), is very preliminary.


## Would some individuals in TMLR's audience be interested in the findings of this paper?

Yes. I think the proposal overall, and the conducted experiments, can inspire useful future work (namely, applying the surrogate GP approach to derive some insight into NNs).


**Broader Impact Concerns:**

No concerns.

**Requested Changes:**

See **"Weaknesses"** - addressing the **Major** point through a paper update or a reply to correct my potential misunderstanding would be most helpful.

**Strengths And Weaknesses:**

## Strengths
- The paper is clear and well-structured.
- The idea of modeling NNs with GPs appears well-motivated and promising.


## Weaknesses

- **Major**: I don't think the paper has a clear, well-validated application presented. Specifically,

	- Sections 4.1 and 4.2 derive insights into NNs using GPs, that can also be derived (and were, in previous papers) from NNs directly (low-frequency bias and depth pathologies). Visually, both left (NNs) and right (GPs) plots of figures 3 and 4 demonstrate the respective phenomena equally convincingly, so there is no clear "value added" from GPs in these examples (to be clear, the authors aren't claiming the opposite).
	- Section 4.3 shows preliminary evidence for NN model selection, but it is only tested on two models where one is, by oracle design, much better than the other. A much more practical setting would be to take real datasets (e.g. UCI, MNIST, CIFAR-10 etc) and compare marginal likelihoods of models obtained during a typical hyper-parameter sweep on them.
	- IIUC, Section 4.4 shows a correlation of generalization error and the GP lengthscale correlation for each fixed hyper-parameter, when sweeping over different datasets. But similarly to the above, isn't a much more practical setting one where we fix datasets, and sweep over hyper-parameters (for the purpose of model selection)? I think measuring these correlations would be much more insightful.

- **Minor**: the question of whether to use infinite-width NN GPs or the proposed surrogate GPs remains open. For example, it could be interesting to see how infinite-width NN GPs do as alternatives to surrogates in sections 4.1, 4.2, and 4.3.2 / 4.3.3 (perhaps they would work despite the small width?).

- **Minor**: I think surrogate GPs should be also considered in section 4.3.1, to show that they apply in the wide setting too (and see how they compare to infinite width GPs).

---

> ### Author Response · Authors · 2022-10-31
> **Author response to Reviewer iVRX**
>
> We thank the reviewer for their time and comments on the submission. We have made a [broader response to all reviewers](https://openreview.net/forum?id=p3pH2EKRQz&noteId=7u6xBExXwk) above, as well as responses to common concerns among reviewers on
> [novelty](https://openreview.net/forum?id=p3pH2EKRQz&noteId=zfC8xqc2Xb),
> [interpretability](https://openreview.net/forum?id=p3pH2EKRQz&noteId=5W9u8WlGme),
> and [practical relevance](https://openreview.net/forum?id=p3pH2EKRQz&noteId=TFdVuqlKtz).
> We also report the [results of an additional experiment](https://openreview.net/forum?id=p3pH2EKRQz&noteId=c6MwSr0uVo) we conducted during the response period. Please let us know either here or in response to those comments whether you have any questions or comments on these aspects.
>
> > ...Section 4.3 shows preliminary evidence for NN model selection, but it is only tested on two models where one is, by oracle design, much better than the other. A much more practical setting would be to take real datasets (e.g. UCI, MNIST, CIFAR-10 etc) and compare marginal likelihoods of models obtained during a typical hyper-parameter sweep on them.
>
> > ...Section 4.4 shows a correlation of generalization error and the GP lengthscale correlation for each fixed hyper-parameter, when sweeping over different datasets. But similarly to the above, isn't a much more practical setting one where we fix datasets, and sweep over hyper-parameters (for the purpose of model selection)? I think measuring these correlations would be much more insightful.
>
> We thank the reviewer for these suggestions. In Section 4.3, we present a subset of model families here to build intuition for later, more systematic results that show benefits of the method for generalization prediction (cf. Section 4.4). As such, we will focus on addressing your comment on having a "typical hyperparameter sweep" in the context of Section 4.4.
>
> For your latter comment, we note that we *do* "sweep" over hyperparameters in a limited sense: We test 4 activation functions and 3 early-stopping iterations (row and column values) to produce the results in Figure 9, as well as datasets (points in each panel). We do think it is valuable to see that the trend persists across datasets, as we report in Figure 9. However, we cannot disagree that "more evidence is better evidence," and so in the response period we have implemented your suggestion of testing a broader set of hyperparameters; we report results in the section of the [main response above](https://openreview.net/forum?id=p3pH2EKRQz&noteId=c6MwSr0uVo).
>
> > Minor: the question of whether to use infinite-width NN GPs or the proposed surrogate GPs remains open. For example, it could be interesting to see how infinite-width NN GPs do as alternatives to surrogates in sections 4.1, 4.2, and 4.3.2 / 4.3.3 (perhaps they would work despite the small width?).
>
> > Minor: I think surrogate GPs should be also considered in section 4.3.1, to show that they apply in the wide setting too (and see how they compare to infinite width GPs).
>
> These are interesting experimental questions that we could not test in the short response period window. We will aim to include these baselines in an updated version of the manuscript.

---

### Review · Reviewer_z2xh · 2022-09-19

**Summary Of Contributions:**

This paper presents a study of neural network predictions with Gaussian process surrogates; the main idea is to gather input-output pairs from a set of neural networks having varying architectures, fit Gaussian process regression models to them, and interpret the results by inspecting the optimized kernel hyperparameters or marginal likelihood values. While simple to implement, the Gaussian process surrogate model successfully captures some interesting properties of neural network predictions, including spectral biases, depth pathologies, and even generalization abilities to some extent.

**Requested Changes:**

- I enjoyed reading the manuscript, the paper is well written and provides some insightful analysis through the proposed framework.
Still, I think the arguments in the paper can be enhanced if the proposed framework can be shown to be scalable or applicable for more generic problems; for instance, would it still work for neural networks for classification problems, taking high-dimensional image data? Inspecting the spectral biases in such cases would be particularly interesting.

- Also, as shown in the lines of work revealing the correspondence between the infinite-width neural networks and neural networks of various architectures, including convolutional neural nets (with residual connections), recurrent neural nets, and even transformers, I wonder if the Gaussian process surrogate would work well for those various architectures (or at least discussion on this matter).

- Seems like the form of the kernels is selected mainly for the reason of explainability; I guess this might limit the flexibility of the GP regression model for more complicated settings. Would there be an alternative kernel that is in the middle ground of interpretability and flexibility?

- How is the number of mixture components $Q$ chosen and how does it affect the analysis?

- I'm confused with the term "data kernel" in section 4.4; the paper states that it is learned directly from the training dataset, then does it mean that you directly fit a GP to the data? If so, why not fitting GP surrogate model for the neural network outputs for the training sets and comparing it to the GP surrogate trained for the validation set?

- In Figure 8, for the wine dataset, the data lengthscales and surrogate lengthscales are different because features 1, 4, 6 are different w.r.t. the threshold lengthscale 1.0, while for the airfoil dataset two lengthscales are similar w.r.t. the threshold 1.0. Why 1.0 for the threshold? Even for the airfoil data the lengthscales are not identical, so if we set different threshold for the lengthscale we might have a different conclusion.

**Strengths And Weaknesses:**

Strength
- The paper is well written and easy to follow.
- Building surrogate models with Gaussian processes is a valid idea.
- The experiments demonstrate interesting use cases of the proposed framework.

Weakness
- The experiments are limited to relatively simple feed-forward neural networks.
- Most of the experiments are done with regression tasks for synthetic target functions and relatively small-scale real datasets.
- Limited practical implication; how can it be useful for real-world applications?

---

> ### Author Response · Authors · 2022-10-31
> **Author response to Reviewer z2xh**
>
> We thank the reviewer for their time and comments on the submission. We have made a [broader response to all reviewers](https://openreview.net/forum?id=p3pH2EKRQz&noteId=7u6xBExXwk) above, as well as responses to common concerns among reviewers on
> [novelty](https://openreview.net/forum?id=p3pH2EKRQz&noteId=zfC8xqc2Xb),
> [interpretability](https://openreview.net/forum?id=p3pH2EKRQz&noteId=5W9u8WlGme),
> and [practical relevance](https://openreview.net/forum?id=p3pH2EKRQz&noteId=TFdVuqlKtz).
> We also report the [results of an additional experiment](https://openreview.net/forum?id=p3pH2EKRQz&noteId=c6MwSr0uVo) we conducted during the response period. Please let us know either here or in response to those comments whether you have any questions or comments on these aspects.
>
> > ...as shown in the lines of work revealing the correspondence between the infinite-width neural networks and neural networks of various architectures, including convolutional neural nets (with residual connections), recurrent neural nets, and even transformers ... \[would\] the Gaussian process surrogate ... work well for that various architecture.
>
> We would like to note that this line of work that the reviewer refers to is several years worth of work across numerous institutions. It is a very high bar indeed to expect this single paper to accomplish something analogous.
>
> > Seems like the form of the kernels is selected mainly for the reason of explainability; I guess this might limit the flexibility of the GP regression model for more complicated settings. Would there be an alternative kernel that is in the middle ground of interpretability and flexibility?
>
> This is an interesting question for future work.
>
> > How is the number of mixture components Q chosen and how does it affect the analysis?
>
> In general, we didn't find that the number of mixture components mattered significantly, and we chose 10 in accordance with other papers in the literature ([Liu et al., *NeurIPS* 2020](https://proceedings.neurips.cc/paper/2020/hash/f52db9f7c0ae7017ee41f63c2a7353bc-Abstract.html)).  We note that the marginal likelihood can prune unnecessary mixture components by setting mixture weights to zero.
>
> > I'm confused with the term "data kernel" in section 4.4; the paper states that it is learned directly from the training dataset, then does it mean that you directly fit a GP to the data? If so, why not fitting GP surrogate model for the neural network outputs for the training sets and comparing it to the GP surrogate trained for the validation set?
>
> The "data kernel" is meant to capture the properties of the dataset *independent of predictions from the neural network population*, and thus we fit the Gaussian process hyperparameters directly to the dataset. The "surrogate kernel", by contrast, is meant to capture the *behavior of a neural network family on the dataset*, and thus we fit the Gaussian process hyperparameters to the behavior of neural networks from that family on that dataset. The intuition here is that, by comparing the data and surrogate kernels, we capture how the neural network family's predictions differ from the ground truth dataset. We will clarify this intuition in the manuscript.

---

### Review · Reviewer_FJ8Z · 2022-10-16

**Summary Of Contributions:**

This paper makes the case of using Gaussian processes (GP) as surrogate models to understand different deep learning phenomena. In particular, the authors argue that fitting a GP to approximate the distribution of functions learned by a neural network can be an effective way to gauge different properties of neural networks and simplify the study of their inductive bias. Several illustrative examples of this approach are provided in which the authors analyze known properties of neural networks, e.g., spectral bias, using their tools on simple MLPs trained to approximate 1D simple targets. Finally, the paper presents some experiments in which the likelihoods of different estimated GP are used to predict generalization on a few low-dimensional regression tasks.

**Broader Impact Concerns:**

I do not believe there exist major impact concerns directly linked to this work.

**Requested Changes:**

Unfortunately, I believe a lot of work would be required for this manuscript to meet the bar of acceptance to TMLR. In particular, I think it is critical that the authors clearly highlight some new insight that their technique can identify which was not possible with previous tools. Similarly, I believe it is fundamental that such technique has the potential to be moderately scaled so that it can be used in practice.

**Strengths And Weaknesses:**

# Strengths
1. **Interesting idea to tackle a topic of great interest:** I find the main premise of this work, i.e., using theoretically grounded surrogate models to understand deep learning, an interesting idea to address the complexity of understanding deep learning. In particular, I believe that the authors make a good case in their introduction in favor of the development of novel tools that can simplify the empirical analysis of the global properties of deep neural networks.
2. **Clear exposition**: Overall, I find the paper to be nicely written and properly self-contained.
3. **Strong effort in reproducibility**: In general, I appreciate the effort made by the authors in making sure that the details of all their experiments are clearly defined (with the exception of which UCI datasets are used in Section 4).

# Weaknesses
1. **Lack of new findings:** The main issue with this work is that it fails to meet its own expectations. In particular, the authors make a strong case in the introduction for the need of developing new tools that allow to better describe and analyze the inductive bias of the neural networks used in practice. Yet, the results presented in this work qualitatively replicate on very toy settings, at most, known facts about neural networks that had been discovered without using these newly developed tools. In this regard, one would expect that the authors show a clear example in which the use of GP surrogate models was key to identify/characterize some deep learning phenomenon which would have been impossible to analyze otherwise.
2. **Scalability issues:** Similarly, another big concern about this work is that the proposed surrogate models suffer from very serious scalability issues (as clearly exemplified by the lack of high-dimensional experiments). In particular, GPs are known to be hard to fit in very high-dimensional settings, and to the best of my knowledge, with our current technology it will be almost impossible to use the proposed techniques on even the most naive setting of describing the distribution of functions learned by a LeNet5 trained on MNIST from different random initializations.
3. **Mostly qualitative results:** On a different note, another weakness of this work is that the conclusions of most of the experiments are based on qualitative and subjective assessments, e.g., Sec. 4.2., or on anecdotal evidence on a few cherry-picked examples, e.g., Fig. 5, and Fig. 6.

---

> ### Author Response · Authors · 2022-10-31
> **Author response to Reviewer FJ8Z**
>
> We thank the reviewer for their time and comments on the submission. We have made a [broader response to all reviewers](https://openreview.net/forum?id=p3pH2EKRQz&noteId=7u6xBExXwk) above, as well as responses to common concerns among reviewers on
> [novelty](https://openreview.net/forum?id=p3pH2EKRQz&noteId=zfC8xqc2Xb),
> [interpretability](https://openreview.net/forum?id=p3pH2EKRQz&noteId=5W9u8WlGme),
> and [practical relevance](https://openreview.net/forum?id=p3pH2EKRQz&noteId=TFdVuqlKtz).
> We also report the [results of an additional experiment](https://openreview.net/forum?id=p3pH2EKRQz&noteId=c6MwSr0uVo) we conducted during the response period. Please let us know either here or in response to those comments whether you have any questions or comments on these aspects.
>
> > **Mostly qualitative results:** ... another weakness of this work is that the conclusions of most of the experiments are based on qualitative and subjective assessments, e.g., Sec. 4.2., or on anecdotal evidence on a few cherry-picked examples, e.g., Fig. 5, and Fig. 6.
>
> We agree with the reviewer that not all experiments in the paper should be weighted equally—we wrote in the initial submission that Sections 4.1 and 4.2 are qualitative sanity-check experiments that serve to validate the method in a previously-established setting. We do not claim that we thoroughly and quantitatively evaluate the method in these sections.
>
> Sections 4.3 and 4.4, which constitute our main "new results" are **quantitative experiments**. We spend the majority of space in the paper on these quantitative experiments.
>
> Figures 5 and 6 in Section 3 display pairwise or three-way comparisons between neural network families. We present a subset of model families here to build intuition for later, more systematic results that show similar benefits of the method for generalization prediction (cf. Section 4.4). The reviewer has not mentioned Figure 7, which broadens the set of neural network families compared over Figure 5 and 6 by including a range of early stopping iterations and activation functions.
>
> Lastly, the reviewer has not mentioned Section 4.4 and Figures 8 and 9, which constitute our most systematic set of experiments across a range *of dimensions of hyperparameter variation*. We have additionally run another quantitative experiment that we describe in the [main response above](https://openreview.net/forum?id=p3pH2EKRQz&noteId=c6MwSr0uVo).

---

> > ### Comment · Reviewer_FJ8Z · 2022-11-07
> > **Reply to authors**
> >
> > Thank you very much for the very detailed answer and compelling rebuttal. However, after having thoroughly reviewed your answers, the new draft and the other reviewer's comments, I still stand by my previous assessment.
> >
> > As the authors mention, there are two main criteria for acceptance to TMLR, and in my opinion this work fails to meet either of them. Let me expand:
> > > [TMLR Criterion #1] Are the claims made in the submission supported by accurate, convincing and clear evidence?
> >
> > The main claim made by this work is that modelling the distribution of learned NN functions using GPs can be a useful tool to understand neural networks. However, nowhere in this draft, there is a convincing example that provides clear evidence that GPs as surrogate models can be used to identify/explain novel phenomena. What is more, the clear lack of scalability of the proposed technique makes it highly unlikely that the proposed technique could ever yield such insights.
> >
> > > [TMLR Criterion #2] Would at least some individuals in TMLR's audience be interested in knowing the findings of this paper?
> >
> > Personally, I understand this criterion as a very soft form of impact quantification where the threshold for acceptance strictily depends on what one defines as the wide TMLR audience. The larger this audience, the lower the bar for acceptance, as certainly there should be at least some individual that is interested in the paper. Unfortunately, deciding what makes the TMLR audience is a subjective guess (thus the need to have multiple reviewers) which makes assessing this criterion much harder. In my opinion, the TMLR audience is a scientifically-literate crowd with broad interests in machine learning and deep learning, interested in an academic discussion of rigorous findings about ML/AI (including both practical and theoretical insights). In this regard, I do not think there are individuals interested in reading the toy results about neural networks presented in this work reiterating known findings using well-established techniques (i.e., GPs).
> >
> > I apologize for being this strict, but I think it is important that as a community we set a high bar for rigorousness and scientific discussions.
> >
> > Mainly for these two reasons, I will stand by my previous assessment, and vote for rejection of this work.

---

> > > ### Author Response · Authors · 2022-11-09
> > > **Second author response to Reviewer FJ8Z (1/2)**
> > >
> > > We thank the reviewer for their further comments during the discussion.  We would like to clarify the scope of our work so that we can come to a clearer assessment with respect to TMLR's Criteria #1 and #2. Here are our claimed contributions, put most simply:
> > >
> > > **(C1)**  We introduce the idea of a "black-box surrogate model" for a population of machine learners (in our particular case, neural networks). We instantiate this idea in a framework that uses a Gaussian process (GP) and marginal likelihood optimization of GP kernel hyperparameters using a behavioral dataset from a population of machine learners.
> > >
> > > **(C2)**  We use this framework to simultaneously do the following:
> > > - identify spectral bias in ReLU networks, as first reported in [Rahaman et al.  (*ICML* 2019)](https://proceedings.mlr.press/v97/rahaman19a.html);
> > > - identify depth pathologies of deep feedforward networks, as first reported in [Duvenaud et al. (*ICML* 2014)](https://proceedings.mlr.press/v33/duvenaud14.html);
> > > - quantitatively predict generalization, including in a setting that is common to the literature on evaluating Gaussian processes and uncertainty calibration (UCI:
> > > [Kuleshov et al., *ICML 2018*](https://proceedings.mlr.press/v80/kuleshov18a.html);
> > > [Pearce et al., *AISTATS* 2020](https://proceedings.mlr.press/v108/pearce20a.html);
> > > [Antoran et al., *NeurIPS* 2020](https://proceedings.neurips.cc/paper/2020/hash/781877bda0783aac5f1cf765c128b437-Abstract.html);
> > > [Kristiadi et al., *UAI* 2021](https://proceedings.mlr.press/v161/kristiadi21a.html);
> > > [Daxberger et al., *ICML* 2021](https://proceedings.mlr.press/v139/daxberger21a.html)).

---

> > > ### Author Response · Authors · 2022-11-09
> > > **Second author response to Reviewer FJ8Z (2/2)**
> > >
> > > ### Regarding TMLR's Criterion #1
> > >
> > > This criterion is in two parts: **(1a)** technical soundness; **(1b)** clarity of the narrative and arguments presented as per [the guide on acceptance criteria](https://www.jmlr.org/tmlr/acceptance-criteria.html).
> > >
> > > **First, we assume there are no concerns with respect to (1b)**, as all reviewers praised the clarity of the manuscript. However, we are happy to hear about any remaining concerns in this vein.
> > >
> > > **Second, regarding (1a)**, the reviewer writes:
> > >
> > > > **there is [not] a convincing example that provides clear evidence that GPs as surrogate models can be used to identify/explain novel phenomena.**
> > >
> > > We do not claim that we capture novel phenomena in the sense of, e.g., a spectral bias or a specific depth pathology. Our contribution here is to show how these phenomena are reflected in a particularly interpretable way in the properties of the estimated GPs. In addition, we show how using GPs as surrogate models can generate meaningful predictions about generalization performance in a setting common to recently published work (see "UCI" above) This is in line with our [author response on novelty](https://openreview.net/forum?id=p3pH2EKRQz&noteId=zfC8xqc2Xb), where we claimed that "we demonstrate that our framework can accommodate **both a qualitative reproduction of several independent prior results at the same time as a prediction of generalization performance**."
> > >
> > > There are a number of published works in machine learning that focus on identifying relationships between different approaches that provide useful insights (rather than introducing "novel phenomena"), and our manuscript is in this genre. Some well-cited examples include:
> > >
> > > Schulman, Heess, Weber, & Abbeel. (2015). Gradient estimation using stochastic computation graphs. *NeurIPS*.
> > >
> > > Gal & Ghahramani. (2016). Dropout as a Bayesian approximation: Representing model uncertainty in deep learning. *ICML*.
> > >
> > > Grant, Finn, Levine, Darrell, & Griffiths. (2018).  Recasting gradient-based meta-learning as hierarchical Bayes. *ICLR*.
> > >
> > > > **the clear lack of scalability of the proposed technique makes it highly unlikely that the proposed technique could ever yield such insights**
> > >
> > > We do not claim to address scalability in this work. However, in addition to classic work on scalable GP approximations (e.g., inducing points:
> > > [Titsias, *AISTATS* 2009](https://proceedings.mlr.press/v5/titsias09a.html)), recent work has made significant progress on scaling exact GP inference to millions of data points
> > > ([Wang et al., *NeurIPS* 2019](https://arxiv.org/abs/1903.08114)) and scaling approximate GP inference to larger datasets like CIFAR-10
> > > ([Sun et al., *ICML* 2021](https://proceedings.mlr.press/v139/sun21d.html));
> > > see [Liu et al., *IEEE Trans* 2020](https://arxiv.org/abs/1807.01065) for a review.
> > > There is no reason in principle that these insights couldn't be applied to our approach.
> > >
> > > However, applying these insights would require significant effort and computational resources. Before committing to such a path, it is important to establish that an approach can produce meaningful results.  We believe that this is what our manuscript does, laying the groundwork to pursue this approach at a larger scale.
> > >
> > > ### Regarding TMLR's Criterion #2
> > >
> > > > **In my opinion, the TMLR audience is a scientifically-literate crowd with broad interests in machine learning and deep learning, interested in an academic discussion of rigorous findings about ML/AI (including both practical and theoretical insights). In this regard, I do not think there are individuals interested in reading the toy results about neural networks presented in this work reiterating known findings using well-established techniques (i.e., GPs).**
> > >
> > > We disagree with your characterization of our results, as summarized above. We believe that our work makes a meaningful contribution to the machine learning literature. Specifically, it shows that the framework we present can reproduce several contemporary insights about the properties of neural networks and has promise as a tool for evaluating when these systems will generalize⁠---something that is relevant to other researchers who may be interested in exploring similar ideas or building on these results.
> > >
> > > > **I think it is important that as a community we set a high bar for rigorousness and scientific discussions.**
> > >
> > > We agree that rigor is valuable to the scientific community. We don't believe that our work lacks rigor. Indeed, it seems like the concerns articulated by the reviewer are primarily about the scope of the claimed contributions rather than rigor. We ask the reviewer to evaluate specifically the claimed contributions (C1) and (C2) identified above and the evidence we provide for these claimed contributions in the manuscript and discussion.

---

### Review · Reviewer_6JqM · 2022-10-23

**Summary Of Contributions:**

This paper argues that Gaussian processes are more interpretable that complicated deep learning "models". Then, it uses GPs as surrogates to try and describe the behaviour and generalisation ranking of deep learning learning models. Some toy examples of this method are investigated with the aim of extracting insight from the learnt GP hyperparameters. Some slightly less toy data is also investigated, where the marginal likelihood of the GP surrogate model is used to describe the generalisation performance of the corresponding neural network model.

**Requested Changes:**

Ideally I would like all of the weakness I listed above fixed or clarified. However, unless I have misunderstood something important in this paper, I do not think the paper is suitable for publication at TMLR even with adjustments.

The central idea seems to be that we can use surrogate GP models to describe the behaviour of neural networks. This idea is not new (see for example, work on infinite width networks), and the experimental evaluation of the approach is not comprehensive enough to shed any new insights onto this idea. The toy datasets considered are not representative enough of situations of practical interest.

Please let me know if I am missing something important.

**Strengths And Weaknesses:**

**Strengths:**
- The paper is very well written. Descriptions are accurate, precise, and cover an appropriate amount of background.


**Weaknesses:**
- Bottom of page 4. I am not convinced that GPs are necessarily interpretable. Having a lengthscale for each dimension does not make a model interpretable. What if the input data were preprocessed via scaling or standardisation? This would change characteristic lengthscales and units without changing the true nature of an underlying data generating mechanic.
- All of the datasets considered before section 4.4 are extremely toy, and I am not sure if any generalisable insight can be gathered from their investigation. For example, the comment "we establish that Gaussian process surrogates reliably rank performance across a range of learning rates and gradient descent algorithms" requires the (very strong) qualification that the result relates to the target function $\sin(05x)$.
- Figure 9. The trend here is not complete enough to take away a strong conclusion. There appears to be some missing factors that change the position of the datapoints in the 2d plane.

Minor:
- "Hyperparameter selection in NNs is not always theoretically grounded." In fact, it is very rarely grounded in anything beyond maximising a coarse objective via a rough search. The following text lists some theoretical results, which are far behind what practitioners are concerned with when choosing hyperparameters for neural nets and are removed from practice.
- Section 4.3.1. Cite Williams for the erf neural net kernel --- 'computing with infinite width neural networks'.
- Is there a reason why the GP surrogates in section 4.4 did not use the corresponding infinite width covariance function as a baseline?

---

> ### Author Response · Authors · 2022-10-30
> **Author response to Reviewer 6JqM**
>
> We thank the reviewer for their time and comments on the submission. We have made a [broader response to all reviewers](https://openreview.net/forum?id=p3pH2EKRQz&noteId=7u6xBExXwk) above, as well as responses to common concerns among reviewers on
> [novelty](https://openreview.net/forum?id=p3pH2EKRQz&noteId=zfC8xqc2Xb),
> [interpretability](https://openreview.net/forum?id=p3pH2EKRQz&noteId=5W9u8WlGme),
> and [practical relevance](https://openreview.net/forum?id=p3pH2EKRQz&noteId=TFdVuqlKtz).
> We also report the [results of an additional experiment](https://openreview.net/forum?id=p3pH2EKRQz&noteId=c6MwSr0uVo) we conducted during the response period. Please let us know either here or in response to those comments whether you have any questions or comments on these aspects.
>
> We respond below to the more specific of the reviewer's comments.
>
> > All of the datasets considered before section 4.4 are extremely toy, and I am not sure if any generalisable insight can be gathered from their investigation. For example, the comment "we establish that Gaussian process surrogates reliably rank performance across a range of learning rates and gradient descent algorithms" requires the (very strong) qualification that the result relates to the target function $\sin(0.5x)$.
>
> This is correct and we will add this qualification (and other necessary qualifications in Sections 4.3) to the paper. We do not intend to overclaim the scope of datasets tested.
>
> > Figure 9. The trend here is not complete enough to take away a strong conclusion. There appears to be some missing factors that change the position of the datapoints in the 2d plane
>
> In addition to the visual representation in Figure 9, we report in each panel a quantitative measure for the inverse relationship between the generalization gap and lengthscale correlation: The Pearson correlation coefficient, ranging from −0.856 to −0.528 across the range of architectures and maximum iteration numbers (panels). We take this < −0.5 correlation as evidence of an inverse relationship between these two variables.
>
> The reviewer notes that there appear to be other factors that mediate the relationship between generalization gap and lengthscale correlation.  If we look closely, this is true for datasets (points) that have suboptimal generalization performance (generalization gap \> 0.1) with the exception of an outlier dataset (green dot in Figure 9 corresponding to the "protein" dataset). In contrast, the lengthscale correlation *strongly* predicts generalization performance for models that generalize well (i.e., there is a tight cluster of points in the bottom right) again with the exception of the same outlier dataset (protein).  (The protein dataset is known to be difficult to model with GPs: See the poor performance of GPs on this dataset in Figure 1 of [Wang et al. (*NeurIPS* 2019)](https://arxiv.org/abs/1903.08114).)
>
> For models with *higher* generalization gap (points outside of the tight cluster of points in the bottom right), it is an *a priori* more difficult prediction problem to exactly predict the generalization gap because there are more ways in which a neural network can be incorrect than correct. It would be interesting to investigate the factors that may explain some variance in the linear relationship between generalization gap and lengthscale correlation for this region; we hypothesize that dataset identity plays a large role here (i.e., datapoints of the same color across panels in most cases maintain their position in Figure 9). We will include such analyses in an updated version of the manuscript; this investigation would be similar to that in Appendix C of [Jiang et al. (*ICLR* 2019)](https://arxiv.org/abs/1810.00113).
>
> Please also see the [additional empirical result](https://openreview.net/forum?id=p3pH2EKRQz&noteId=c6MwSr0uVo) we present above in the main comment.
>
> > Minor: "Hyperparameter selection in NNs is not always theoretically grounded." In fact, it is very rarely grounded in anything beyond maximising a coarse objective via a rough search. The following text lists some theoretical results, which are far behind what practitioners are concerned with when choosing hyperparameters for neural nets and are removed from practice.
>
> We agree that there is a gap between the theory and practice of hyperparameter selection for deep learning; we do not intend to claim that one does not exist in this work. We will clarify this in the manuscript.
>
> > Is there a reason why the GP surrogates in section 4.4 did not use the corresponding infinite width covariance function as a baseline?
>
> In this experiment, we aimed to focus on whether the data-driven kernel can reliably predict generalization performance. The infinite-width approximation is an appropriate baseline—we were not able to include it as such during the review period, but will include this baseline in an updated version of the manuscript.

---

### Author Response · Authors · 2022-10-30
**Author response to all reviewers**

We thank the reviewers for their comments. To summarize what we gathered from the comments, the reviewers thought the submission was well-written and precise, the idea was well-motivated and promising, and we made a good effort towards reproducibility. (On the point of reproducibility, we are in the process of preparing for release our code repository with documentation, scripts, and intermediate results that are sufficient to reproduce all results reported in the submission.)

We understand that the main concerns with the submission appear to be about **practical applicability of the method**—that is, that we do not demonstrate that the method we explore in this work can be scaled to the large-scale neural network models that are commonly used in practice. On this point, we disagree with the reviewers that a scaled demonstration is a necessity for publication at TMLR; we provide more details in a response specifically targeting "practical relevance."

Our goal in this response is work with the reviewers to understand their assessment with respect to the two reviewing criteria at TMLR; from the [*TMLR guidelines on "Evaluation criteria"*](https://jmlr.org/tmlr/reviewer-guide.html#h.q2ee6w3zmhbr):

> The acceptance decision for a submission is based on the answers to the following questions:
> [**TMLR Criterion #1**] Are the claims made in the submission supported by accurate, convincing and clear evidence?
> [**TMLR Criterion #2**] Would at least some individuals in TMLR's audience be interested in knowing the findings of this paper?
> Papers should be accepted if they meet the criteria, even if the contribution or significance of the work is modest.

Because several reviewers already noted that the idea is well-motivated and interesting, we assume that there are no major concerns with TMLR Criterion #2 (but please let us know if otherwise). We respond below to shared comments of the reviewers in order to address concerns surrounding TMLR Criteria #1; our goal is to come to a shared agreement on what the claims of the submission should be in the context of the results we present in the paper (as well as the additional experiment we present below). We note the adjustments covered by the TMLR review process; from the [*TMLR guidelines on "Evaluation criteria"*](https://jmlr.org/tmlr/reviewer-guide.html#h.q2ee6w3zmhbr):

> Any gap between claims and evidence should be addressed by the authors.  Often, this will lead reviewers to ask the authors to provide more evidence by running more experiments. However, this is not the only way to address such concerns. Another is simply for the authors to adjust (reduce) their claims.

We plan to make changes in both of these respects: In addition to modifying the scope of our claims in line with the reviewers' requests, we have conducted an additional experiment as detailed in a follow-up response.

We thank the reviewers for their continued comments and suggestions.

---

> ### Author Response · Authors · 2022-10-30
> **Author response to all reviewers: New experimental result**
>
> Based on the suggestion of Reviewer iVRX, during the response period we have run an additional experiment to augment the results reported in Figure 9 of the submitted manuscript. In Figure 9, we correlated generalization gap and lengthscale correlation for 3 × 4 = 12 neural network families (corresponding to 3 early stopping iterations and 4 activation function) and across UCI datasets, and demonstrated that there was an inverse relationship between these two quantities; i.e., high lengthscale correlation predicted low generalization gap.
>
> To augment this, as suggested by Reviewer iVRX, we conducted a broader hyperparameter sweep across activation functions, learning rates, neural network widths and neural network depths for a total of 32 hyperparameter settings; the goal of this experiment is to simulate a more standard model selection process in which the goal is to be able to select an optimal hyperparameter setting among many possible options. We find that the inverse relationship we demonstrated in the more restricted model selection process persists in the majority of UCI datasets we tested (i.e., that high lengthscale correlation predicts low generalization gap), with the exception of two datasets for which we give a preliminary explanation in the caption.
>
> This model selection process exemplifies a way in which training a surrogate model (the Gaussian process) to approximate a family of neural networks can be used to anticipate the consequences of conducting a hyperparameter sweep, illustrating the potential of this approach. Please see the expanded results in the updated revision of the manuscript in **Section 4.5**.

---

> ### Author Response · Authors · 2022-10-30
> **Author response to all reviewers: On novelty**
>
> ### To Reviewer 6JqM:
>
> > The central idea seems to be that we can use surrogate GP models to describe the behaviour of neural networks. This idea is not new (see for example, work on infinite width networks) ...
>
> We do not claim that we are the first to use GPs to describe the behavior of neural networks.
> We explore the idea of **black-box surrogates** for neural networks, as defined in the introduction (paragraph 4) and the background section (paragraph 1). As we noted there, this terminology is borrowed from engineering design, and describes a modeling process in which the internal details of a system are abstracted away and only the input-output behavior of the system is analyzed.
>
> This idea is related to but **distinct** from the approaches the reviewer refers to ("infinite width networks"), and we addressed this point in the introduction of the submission:
>
> > Gaussian processes (GPs) are a natural choice, with appealing theoretical properties specific to the study of neural networks (NNs); namely, certain limiting cases of NN architectures are realizable as GPs (Neal 1996; Li and Y. Liang 2018; Jacot et al. 2018; Allen-Zhu et al. 2019; Du et al. 2019). However, in contrast to these analytic approaches, we aim to explore the scientific and practical utility of idealizing NNs with GPs using a data-driven approach to estimating the kernel functions.
>
> The kernels derived using the "infinite-width limit" of neural networks are the result of a mathematical derivation made with specific architectural assumptions in an asymptotic regime. For instance, computing the kernel using the relationships derived by [Lee et al.  (2018)](https://arxiv.org/abs/1711.00165) involves solving the Gaussian integral in Eq. (4) derived on page 4, which depends on the infinite-width assumption and other details about the architecture.
>
> In contrast, we **estimate kernel functions using input-output behavior of a population of neural networks** (which we referred to in the submission as the "data-driven approach to estimating ... kernel functions"). This approach does not make the infinite-width assumption and is agnostic to details of the architecture. We think that the data-driven approach to estimating kernel functions has potential value as a means of understanding neural network models for exactly this reason (relaxing assumptions about the architectures under study).
>
> In this submission, we conduct a preliminary study to demonstrate that the data-driven approach to estimating kernels makes sense for a preliminary set of neural network models, on tasks that have been used in qualitative studies of neural network behavior as well as on benchmarks that are commonly used in evaluating Gaussian process models (for examples, see [*Benton et al.  (2019)*](https://arxiv.org/abs/1910.13565); [*Hamid et al.  (2022)*](https://arxiv.org/abs/2106.07452)). We think that at least some of TMLR's audience would be interested in knowing about the explorations we conduct in this paper, which is the appropriate criterion for evaluation as per [*TMLR's reviewing guidelines*](https://jmlr.org/tmlr/reviewer-guide.html#h.q2ee6w3zmhbr), instead of "novelty."
>
> ### To Reviewer FJ8Z:
>
> > **Lack of new findings:** ... the results presented in this work qualitatively replicate on very toy settings, at most, known facts about neural networks that had been discovered without using these newly developed tools
>
> > ...it is critical that the authors clearly highlight some new insight that their technique can identify which was not possible with previous tools.
>
> We explicitly state in the submission at the beginning of Section 4, "Experiments," that "In Sections (4.1) and (4.2), we capture previously established NN phenomena." In these instances, we do not intend to introduce novel phenomena.
>
> In contrast, "...in Sections (4.3) and (4.4), we predict NN generalization behavior." We would not describe this as a "fact about neural network" behavior but instead a **quantitative prediction of a neural network's extrapolation behavior**, which is certainly of interest to the machine learning community (see the large amount of work in model selection and generalization prediction; e.g., [Jiang et al. (2019)](https://arxiv.org/abs/1912.02178); [Jiang et al. (2020)](https://arxiv.org/abs/2012.07976); [Lotfi et al. (2022)](https://arxiv.org/abs/2202.11678).
>
> Our main contribution is that we demonstrate that our framework can accommodate **both a qualitative reproduction of several independent prior results at the same time as a prediction of generalization performance**.

---

> ### Author Response · Authors · 2022-10-30
> **Author response to all reviewers: On interpretability of Gaussian processes**
>
> ### From Reviewer 6JqM:
>
> > Bottom of page 4. I am not convinced that GPs are necessarily interpretable. Having a lengthscale for each dimension does not make a model interpretable. What if the input data were preprocessed via scaling or standardisation? This would change characteristic lengthscales and units without changing the true nature of an underlying data generating mechanic.
>
> ### From Reviewer z2xh:
>
> > In Figure 8, for the wine dataset, the data lengthscales and surrogate lengthscales are different because features 1, 4, 6 are different w.r.t. the threshold lengthscale 1.0, while for the airfoil dataset two lengthscales are similar w.r.t. the threshold 1.0. Why 1.0 for the threshold? Even for the airfoil data the lengthscales are not identical, so if we set different threshold for the lengthscale we might have a different conclusion.
>
> First, **we note that we preprocessed the input data** so that each feature dimension has mean zero and standard deviation one. Since the range of all features is in \[-1, 1\] by this standardization process, a dimension with a lengthscale much greater than 1 can be interpreted as an unimportant dimension, since it does not contribute greatly to the covariance function value.
>
> Second, we note that **the *****relative***** feature importances as interpreted from the lengthscales are meaningfu**l. Indeed, changing the scale of the data does change the absolute value of the learned lengthscales. However, the lengthscales retain a clear interpretation with respect to the *scaled* units; in particular, the lengthscale determines the distance along a particular input dimension to expect meaningful correlation between output values.
>
> This does all assume a canonical representation of the input data; i.e., that input dimensions have a privileged basis in which their identity is meaningful. However, this assumption is common to various methods that perform importance analyses via saliency or statistical influence, e.g.,
> [Selvaraju et al. (*CVPR* 2017)](https://arxiv.org/abs/1610.02391);
> [Koh & Liang (*ICML* 2017)](https://arxiv.org/abs/1703.04730).
>
> However, we do not claim (nor do we intend) to give guarantees in the presence of arbitrary transformations of the input representation.  Feature-wise standardization, as the reviewer suggests and as we perform, would change the *absolute* values of the lengthscales, but their relative magnitudes would remain meaningful; such rescaling would affect the importance analyses mentioned above in a similar manner.
>
> Third, and regarding the interpretability of GPs, **there is a well-established precedent for using Gaussian processes as interpretable models** in applied disciplines such as environmental, medical sciences, and astronomy; for example:
> [Duvenaud et al. (*ICML* 2013)](https://doi.org/10.48550/arXiv.1302.4922);
> [Miller et al. (*NeurIPS* 2015)](https://papers.nips.cc/paper/2015/hash/7fb8ceb3bd59c7956b1df66729296a4c-Abstract.html);
> [Colopy et al. (*EMBC* 2016)](https://doi.org/10.1109/EMBC.2016.7591926);
> [Camps-Valls et al. (*National Science Review* 2019)](https://doi.org/10.1093/nsr/nwz028);
> [Cheng et al. (*Nature Communications* 2019)](https://doi.org/10.1038/s41467-019-09785-8);
> [Cheng et al. (*BMC Med Inform Decis Mak* 2020)](https://doi.org/10.1186/s12911-020-1069-4);
> [Steinruecken et al. (*Automated Machine Learning* 2019)](HTTPS://DOI.ORG/10.1007/978-3-030-05318-5_9).

---

> ### Author Response · Authors · 2022-10-30
> **Author response to all reviewers: On practical relevance**
>
> ### From Reviewer 6JqM:
>
> > ... the experimental evaluation of the approach is not comprehensive enough to shed any new insights onto this idea. The toy datasets considered are not representative enough of situations of practical interest.
>
> ### From Reviewer FJ8Z:
>
> > ... another big concern about this work is that the proposed surrogate models suffer from very serious scalability issues (as clearly exemplified by the lack of high-dimensional experiments). In particular, GPs are known to be hard to fit in very high-dimensional settings, and to the best of my knowledge, with our current technology it will be almost impossible to use the proposed techniques on even the most naive setting of describing the distribution of functions learned by a LeNet5 trained on MNIST from different random initializations.
>
> > ... I believe it is fundamental that such technique has the potential to be moderately scaled so that it can be used in practice.
>
> ### From Reviewer z2xh:
>
> > The experiments are limited to relatively simple feed-forward neural networks.
>
> > Most of the experiments are done with regression tasks for synthetic target functions and relatively small-scale real datasets.
>
> > Limited practical implication; how can it be useful for real-world applications?
>
> ### From Reviewer iVRX:
>
> > I don't think the paper has a clear, well-validated application presented.
>
> Broadly, we believe that the reviewers are concerned that the method presented in the paper does not have a "killer app(lication)" that will be of broad practical interest. We do not claim to have a killer app in this first presentation of this method; instead, we claim that we **conduct a preliminary study to demonstrate that the data-driven approach to estimating kernels produces reasonable qualitative and qualitative results for a restricted set of neural network models**.
>
> The reason that we selected TMLR as a venue is that it encourages exploratory work without "perceived impact" as a bar for acceptance.  From the [*TMLR guidelines*](https://jmlr.org/tmlr/reviewer-guide.html#h.q2ee6w3zmhbr):
>
> > Crucially, the criterion of whether a paper is interesting to the TMLR audience should not be used as a reason to reject work that isn't considered "significant" or "impactful" because it isn't achieving a new state-of-the-art on some benchmark. ... We explicitly avoid these terms ("significant", "impactful", "novel"), and focus instead on the notion of "interest". If the authors make it clear that there is something to be learned by some researchers in their area from their work, then the criteria of interest is considered satisfied.
>
> As such, we do not intend to claim, nor do we claim, that we demonstrate "real world," "practical," "scaled" impact, and as per the criterion above, we are not required to do so in this submission to meet the bar for acceptance at TMLR. We do believe that there is **potential** for such things, as we wrote in the Discussion section of the submission:
>
> > First, though the framework is in principle applicable to broader settings, we restricted this first exploration to regression tasks and feed-forward neural network architectures. A broader study of more architectures on more types of tasks would be challenging due to the need to scale Gaussian processes but potentially rewarding, as characterizing properties of neural networks as used in practice is a significant open problem with far-reaching implications (Sejnowski 2020).
>
> We will make this statement more clearly in the introduction, alongside the broader motivation of interpretability for neural networks.

---

### Decision · Action_Editors · 2022-12-02

**Recommendation:** Reject

**Comment:**

This is a difficult decision. I think that the framework is well motivated, and the paper is written clearly. The main issue that the reviewers have, and that I share, is that it is currently a very large leap to go from the toy experiments in this paper to real networks on practical problems - one that may not be possible at all (sparse GPs are not trivial to fit properly compared to a standard GP).

The claims of the paper can be scaled back to "A framework for analyzing neural networks on toy problems using GPs," but then this framing substantially reduces the audience that may find it interesting and valuable. This creates a tension between acceptance criteria 1 and 2.

The authors mention that large-scale experiments would be far more resource intensive and would constitute a paper on its own. I don't disagree. However, I do think there are two possible paths forward:

1) Present synthetic examples using sparse GPs that successfully reproduce the results of the full GP. This is far less intensive than large-scale experiments on real datasets and would provide some signal that the framework is scalable.

2) Look into experiments from the recent literature on neural networks for scene representation, e.g., NeRF models (Mildenhall et al., ECCV 2020). These are generally far smaller than image classification models, operate in low-dimensional spaces, and are of practical usage today. This is also an area where the phenomenon of spectral bias is especially important, so insights of this sort are more likely to be of interest.

In summary, to fit the current claims and to be of sufficient interest to the community, I think that the paper needs a stronger signal that the framework is practical and/or scalable. It certainly doesn't need to go all the way towards large parameter models or anything like that, but just a bit more than what is currently offered would put it over the bar.

**Audience:**

Yes, I do believe that some researchers would be interested. The paper is clearly written and does recover some phenomena that had been previously discovered in the literature. However, the limitation to toy experiments greatly diminishes this factor.

**Claims And Evidence:**

Not entirely. The paper develops a framework for using GPs as surrogate models for neural networks with the goal of interpretability. It accomplishes this, but only for a very narrow set of simple tasks. In the view of the majority of the reviewers, this does not constitute convincing evidence.

E.g., the paper states "Taken together, these results suggest that Gaussian process surrogates may be a valuable empirical tool for investigating deep learning, and future work could aim to use this framework to complement existing approaches to interpretability (e.g., Ribeiro et al. 2016) and extrapolation (e.g., Xu et al. 2021)."

This would be the case if there was a more practical experiment, otherwise I think this statement would have to be qualified that GP surrogates may be a useful tool in small-scale settings, but that its value in larger and more realistic settings remains to be seen.